# SGD WITH LARGE STEP SIZES LEARNS SPARSE FEATURES

## ABSTRACT

We showcase important features of the dynamics of the Stochastic Gradient Descent (SGD) in the training of neural networks. We present empirical observations that commonly used large step sizes (i) lead the iterates to jump from one side of a valley to the other causing *loss stabilization*, and (ii) this stabilization induces a hidden stochastic dynamics that *biases it implicitly* toward simple predictors. Furthermore, we show empirically that the longer large step sizes keep SGD high in the loss landscape valleys, the better the implicit regularization can operate and find sparse representations. Notably, no explicit regularization is used so that the regularization effect comes solely from the SGD dynamics influenced by the step size schedule. Therefore, these observations unveil how, through the step size schedules, both gradient and noise drive together the SGD dynamics through the loss landscape of neural networks. We justify these findings theoretically through the study of simple neural network models as well as qualitative arguments inspired from stochastic processes. Finally, this analysis allows to shed a new light on some common practice and observed phenomena when training deep networks.

## 1 INTRODUCTION

Deep neural networks have accomplished remarkable achievements on a wide variety of tasks. Yet, the understanding of their remarkable effectiveness remains incomplete. From an optimization perspective, stochastic training procedures challenge many insights drawn from convex models. E.g., large step-size schedules used in practice lead to unexpected patterns of stabilizations and sudden drops in the training loss, see e.g. He et al. (2016). From a generalization perspective, overparametrized deep nets generalize well while fitting perfectly the data and without any explicit regularizers (Zhang et al., 2017). This suggests that optimization and generalization are tightly intertwined: neural networks find solutions that generalize well *thanks* to the optimization procedure used to train them. This property, known as *implicit bias* or *algorithmic regularization*, has been studied recently both for regression (Li et al., 2018; Woodworth et al., 2020) and classification (Soudry et al., 2018; Lyu and Li, 2020; Chizat and Bach, 2020). However, for all these theoretical results, it is also shown that typical timescales needed to enter the beneficial feature learning regimes are prohibitively long (Woodworth et al., 2020; Moroshko et al., 2020).

In this paper, we aim at staying closer to the experimental practice and consider the SGD schedules from the ResNet paper (He et al., 2016) where the *large step size* is first kept constant and then decayed, potentially multiple times. We illustrate this behavior in Fig. 1 where we reproduce a minimal setting without data augmentation or momentum, and with only one step size decrease. We draw attention to two key observations regarding the large step-size phase: (a) quickly after the start of training, the loss remains approximately constant on average and (b) despite no progress on the training loss, running this phase for longer leads to better generalization. We refer to such large step-size phase as *loss stabilization*. The better generalization hints at some *hidden dynamics* in the parameter space not captured by the loss curves in Fig. 1. Our main contribution is to unveil the hidden dynamics behind this phase: loss stabilization helps to amplify the noise of SGD that drives the network towards a solution with *sparser features* (see Appendix, Figure 7 for a 2D-visualization).

### 1.1 OUR CONTRIBUTIONS

**The effective dynamics behind loss stabilization.** We characterize two main components of the SGD dynamics with large step sizes: (i) a fast movement determined by the bouncing directions

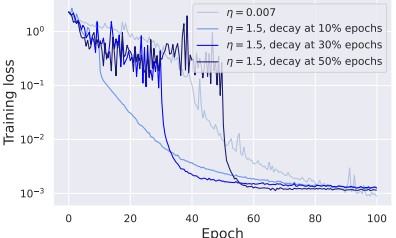 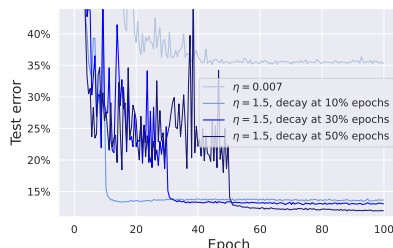

**Figure 1: A typical training dynamics for a ResNet-18 trained on CIFAR-10.** We use weight decay but no momentum or data augmentation for this experiment. We see a substantial difference in generalization (as large as 12% vs. 35% test error) depending on the step size $\eta$ and its schedule. When the training loss stabilizes, there is a hidden progress occurring which we aim to characterize.

causing loss stabilization, (ii) a slow dynamics driven by the combination of the gradient and the multiplicative noise—which is non-vanishing due to the loss stabilization.

**SDE model and sparse feature learning.** We model the *effective* slow dynamics during loss stabilization by a stochastic differential equation (SDE) whose multiplicative noise is related to the neural tangent kernel features, and validate this modeling experimentally. Building on the existing theory on diagonal linear networks, which shows that this noise structure leads to sparse predictors, we conjecture a similar "sparsifying" effect on the features of more complex architectures. We experimentally confirm this on neural networks of increasing complexity.

**Insights from our understanding.** We draw a clear general picture: the hidden optimization dynamics induced by large step sizes and loss stabilization enable the transition to a sparse feature learning regime. We argue that after a short initial phase of training, SGD *first* identifies sparse features of the training data and eventually fits the data when the step size is decreased. Finally, we discuss informally how many deep learning regularization methods (weight decay, BatchNorm, SAM) may also fit into the same picture.

## 1.2 RELATED WORK

He et al. (2016) popularized the piecewise constant step-size schedule which often exhibits a clear loss stabilization pattern. However, they did not provide any explanations for such training dynamics and its implicit regularization effect. Non-monotonic patterns of the training loss have been explored in recent works. However, the loss stabilization regime we consider is different (i) from the catapult mechanism (Lewkowycz et al., 2020) where the training loss shows only one spike at the start of training and then monotonically converges without stabilization, and (ii) from the edge of stability regime of full-batch GD (Cohen et al., 2021) where the training loss shows many regular spikes after some point in training but again without stabilization.

Past works conjectured that large step sizes induce the minimization of some hidden complexity measures related to flatness of minima (Keskar et al., 2016; Smith and Le, 2018). Notably, Xing et al. (2018) point out that SGD moves through the loss landscape bouncing between the walls of a valley where the role of the step size is to guide the noisy iterates of SGD towards a flatter minimum. However, many typically used flatness definitions are questionable for this purpose since (1) they are not invariant under reparametrizations that lead to an equivalent neural network (Dinh et al., 2017), and (2) even for naturally trained networks, full-batch gradient descent with large step sizes (unlike SGD) can lead to flat solutions which are not well-generalizing (Kaur et al., 2022). Note that it is possible to bridge the gap between GD and SGD by using explicit regularization as in Geiping et al. (2022). We instead focus on the *implicit* regularization of SGD which remains the most practical approach for training deep networks.

The importance of large step sizes has been investigated with diverse motivations. However, we believe that existing approaches do not sufficiently capture the *hidden stochastic dynamics* behind the loss stabilization phenomenon observed for deep networks. Attempts to explain it on strongly convex models (Nakkiran, 2020; Wu et al., 2021; Beugnot et al., 2022) are inherently incomplete since it is a phenomenon related to the existence of many zero solutions with very different generalization

properties. Li et al. (2019b) analyzed the role of loss stabilization for a synthetic distribution containing different patterns, but it is not clear how this analysis can be extended to general problems. Works based on stability analysis characterize the properties of the minimum that SGD or GD can potentially converge depending on the step size (Wu et al., 2018; Mulayoff et al., 2021; Ma and Ying, 2021; Nacson et al., 2022). However, these approaches *do not capture the entire training dynamics such as the large step size phase that we consider where SGD converges only after the step size is decayed.* SGD with label noise has been studied because of its beneficial regularization effect and its resemblance to SGD's standard noise. Its implicit bias has been first characterized by Blanc et al. (2020) and extended by Li et al. (2022). However, their analysis only holds in the final phase of the training, close to a zero-loss manifold. Our work instead is closer in spirit to Pillaud-Vivien et al. (2022) where the label noise dynamics is analyzed in the *central* phase of the training, *i.e., when the training loss is still substantially above zero.*

## 2 THE EFFECTIVE DYNAMICS OF SGD WITH LARGE STEP-SIZE: SPARSE FEATURE LEARNING

In this section, we show that large step sizes lead the loss to stabilize by making SGD bounce above a valley. We then unveil the effective dynamics induced by this loss stabilization. To clarify our exposition we showcase our results for the mean square error but other losses like the cross-entropy carry the same key properties in terms of the noise covariance (Wojtowytsch, 2021b, Lemma 2.14). We consider a generic parameterized family of prediction functions $\mathcal{H} := \{x \to h_\theta(x), \theta \in \mathbb{R}^p\}$, a setting which encompasses neural networks. In this case, the training loss on input/output samples $(x_i, y_i)_{1 \le i \le n} \in \mathbb{R}^d \times \mathbb{R}$ reads

$$\mathcal{L}(\theta) := \frac{1}{2n} \sum_{i=1}^n (h_\theta(x_i) - y_i)^2. \tag{1}$$

We consider the overparameterized setting, i.e. $p \gg n$, hence, there shall exists many parameters $\theta^*$ that lead to zero loss, i.e., perfectly interpolate the dataset. Therefore, the question of which interpolator the algorithm converges to is of paramount importance in terms of generalization. We focus on the SGD recursion with step size $\eta > 0$, initialized at $\theta_0 \in \mathbb{R}^p$: for all $t \in \mathbb{N}$,

$$\theta_{t+1} = \theta_t - \eta(h_{\theta_t}(x_{i_t}) - y_{i_t}) \nabla_\theta h_{\theta_t}(x_{i_t}), \tag{2}$$

where $i_t \sim \mathcal{U}(\llbracket 1, n \rrbracket)$ is the uniform distribution over the sample indexes. In the following, note that SGD with mini batches of size $B > 1$ would lead to similar analysis but with $\eta/B$ instead of $\eta$.

### 2.1 BACKGROUND: SGD IS GD WITH SPECIFIC LABEL NOISE

To emphasize the combined roles of gradient and noise, we highlight the connection between the SGD dynamics and that of full-batch GD plus a specific label noise. Such manner of reformulating the dynamics has already been used in previous works attempting to understand the specificity of the SGD noise (HaoChen et al., 2021; Ziyin et al., 2022). We formalize it in the following proposition.

**Proposition 1.** *Let $(\theta_t)_{t \ge 0}$ follow the SGD dynamics Eq.(2) with sampling function $(i_t)_{t \ge 0}$. Let $\mathbf{1}_{i=i_t}$ be indicator function, define for $t \ge 0$, the random vector $\xi_t \in \mathbb{R}^n$ such that for all $i \in \llbracket 1, n \rrbracket$,*

$$[\xi_t]_i := (h_{\theta_t}(x_i) - y_i)(1 - n\mathbf{1}_{i=i_t}). \tag{3}$$

*Then $(\theta_t)_{t \ge 0}$ follows the full-batch gradient dynamics on $\mathcal{L}$ with label noise $(\xi_t)_{t \ge 0}$, that is*

$$\theta_{t+1} = \theta_t - \frac{\eta}{n} \sum_{i=1}^n (h_{\theta_t}(x_i) - y_i^t) \nabla_\theta h_{\theta_t}(x_i), \tag{4}$$

*where we define the random labels $y^t := y + \xi_t$. Furthermore, $\xi_t$ is a mean zero random vector with variance such that $\frac{1}{n(n-1)} \mathbb{E} \|\xi_t\|^2 = 2\mathcal{L}(\theta_t)$.*

This reformulation shows two crucial aspects of the SGD noise: (i) the noisy part at state $\theta$ always belongs to the linear space spanned by $\{\nabla_\theta h_\theta(x_1), \ldots, \nabla_\theta h_\theta(x_n)\}$, and (ii) it scales as the training loss. Concerning (ii), we highlight in the following section that the loss can stabilize because of large step sizes which makes the effective scale of label noise constant. These features are of paramount importance when modelling the effective dynamics that take place during loss stabilization.

## 2.2 THE EFFECTIVE DYNAMICS BEHIND LOSS STABILIZATION

**On loss stabilization.** For generic quadratic costs, e.g., $F(\beta) := \|X\beta - y\|^2$, gradient descent with step size $\eta$ is convergent for $\eta < 2/\lambda_{\max}$, divergent for $\eta > 2/\lambda_{\max}$ and converges to a bouncing 2-periodic dynamics for $\eta = 2/\lambda_{\max}$, where $\lambda_{\max}$ is the largest eigenvalue of the Hessian. However, the practitioner is not likely to hit perfectly this unstable step size and, almost surely, the dynamics shall either converge or diverge. Yet, non-quadratic costs bring to this picture a particular complexity: it has been shown that, even for non-convex toy models, there exist an open interval of step sizes for which the gradient descent neither converge nor diverge (Ma et al., 2022; Chen and Bruna, 2022). As we are interested in SGD, we complement this result by presenting a toy example in which loss stabilization occurs almost surely *in the case of stochastic updates*. Indeed, consider a regression problem with quadratic parameterization on one-dimensional data inputs $x_i$'s, coming from a distribution $\hat{\rho}$, and outputs generated by the linear model $y_i = x_i \theta_*^2$. The loss writes $F(\theta) := \frac{1}{4} \mathbb{E}_{\hat{\rho}} \left( y - x\theta^2 \right)^2$, and the SGD iterates with step size $\eta > 0$ follow, for any $t \in \mathbb{N}$,

$$\theta_{t+1} = \theta_t + \eta \, \theta_t \, x_{i_t} \left( y_{i_t} - x_{i_t} \theta_t^2 \right) \qquad \text{where} \quad x_{i_t} \sim \hat{\rho}. \tag{5}$$

For the sake of concreteness and clarity, suppose that $\theta_* = 1$ and $\operatorname{supp}(\hat{\rho}) = [a, b]$, we have the following proposition (a more general result can be found in Proposition 3 of the Appendix).

**Proposition 2.** *For any $\eta \in (a^{-2}, 1.25 \cdot b^{-2})$ and initialization $\theta_0 \in (0, 1)$, for $t > 0$,*

$$\delta_1 < F(\theta_t) < \delta_2 \qquad \text{almost surely, and} \tag{6}$$

$$\exists T > 0, \forall k > T, \qquad \theta_{t+2k} < 1 < \theta_{t+2k+1} \qquad \text{almost surely.} \tag{7}$$

*where $\delta_1, \delta_2, T > 0$ are constant given in the Appendix.*

The proposition is divided in two parts: if the step size is large enough, Eq.(6) the loss stabilizes in between level sets $\delta_1$ and $\delta_2$ and Eq.(7) shows that after some initial phase, the iterates bounce from one side of the *loss valley* to the other one. Note that despite the stochasticity of the procedure, the results hold *almost surely*.

**The effective dynamics.** As observed in the prototypical SGD training dynamics of Fig. 1 and proved in the non-convex toy model of Proposition 2, large step sizes lead the loss to stabilize around some level set. To further understand the effect of this loss stabilization in parameter space, we shall assume perfect stabilization. Then, from Proposition 1, we conjecture the following behaviour

*During loss stabilization, SGD is well modelled by GD with constant label noise.*

Label noise dynamics have been studied recently (Blanc et al., 2020; Damian et al., 2021; Li et al., 2022) thanks to their connection with Stochastic Differential Equations (SDEs). To properly write a SDE model, the drift should match the gradient descent and the noise should have the correct covariance structure (Li et al., 2019a; Wojtowytsch, 2021a). Proposition 1 implies that the noise at state $\theta$ is spanned by the gradient vectors $\{\nabla_\theta h_\theta(x_1), \dots, \nabla_\theta h_\theta(x_n)\}$ and has a constant intensity corresponding to the loss stabilization at a level $\delta > 0$. Hence, we propose the following SDE model

$$d\theta_t = -\nabla_\theta \mathcal{L}(\theta_t) dt + \sqrt{\eta \delta} \, \phi_{\theta_t}(X)^\top dB_t, \tag{8}$$

where $(B_t)_{t \geq 0}$ is a standard Brownian motion in $\mathbb{R}^n$ and $\phi_\theta(X) := [\nabla_\theta h_\theta(x_i)^\top]_{i=1}^n \in \mathbb{R}^{n \times p}$ referred to as the *Neural Tangent Kernel (NTK) feature matrix* (Jacot et al., 2018). This SDE can be seen as *the effective slow dynamics* that drives the iterates while they bounce *rapidly* in some directions at the level set $\delta$ (fast dynamics). It highlights the combination of the deterministic part of the full-batch gradient and the noise induced by SGD at level set $\delta$ which depends on the step size of SGD. We confirm the validity of this SDE modeling empirically in Sec. C showing that the SDE captures the dynamics of large step size SGD even for non-linear networks. In the next section, we leverage the SDE (8) to understand the implicit bias of such learning dynamics.

## 2.3 SPARSE FEATURE LEARNING

In this section, we give insights on the effective dynamics given by Eq.(8). We begin with a simple model of diagonal linear networks that showcase a sparsity inducing dynamics and further disclose our general message about the overall implicit bias promoted by the effective dynamics.

### 2.3.1   A WARM-UP: DIAGONAL LINEAR NETWORKS

An appealing example of simple non-linear networks that help in forging an intuition for more complicated architectures is diagonal linear networks (Vaskevicius et al., 2019; Woodworth et al., 2020; HaoChen et al., 2021; Pesme et al., 2021). They are two-layer linear networks with only diagonal connections: the prediction function writes $h_{u,v}(x) = \langle u, v \odot x \rangle = \langle u \odot v, x \rangle$ where $\odot$ denotes *elementwise* multiplication. Even though the loss is convex in the associated linear predictor $\beta := u \odot v \in \mathbb{R}^d$, it is not in $(u, v)$, hence the training of such simple models already exhibit a rich non-convex dynamics. In this case, $\nabla_u h_{u,v}(x) = v \odot x$, and the SDE model Eq.(8) writes

$$\mathrm{d}u_t = -\frac{1}{n}\left[X^\top(X(u_t \odot v_t) - y)\right] \odot v_t\,\mathrm{d}t + \sqrt{\eta\delta}\,v_t \odot \left[X^\top \mathrm{d}B_t\right], \qquad (9)$$

where $(B_t)_{t\geq 0}$ is a standard Brownian motion in $\mathbb{R}^n$. Equations are symmetric for $(v_t)_{t\geq 0}$.

**What is the behaviour of this effective dynamics?** Pillaud-Vivien et al. (2022) answered this question by analyzing a similar stochastic dynamics and unveiled the sparse nature of the resulting solutions. Indeed, under sparse recovery assumptions, denoting $\beta^*$ the sparsest linear predictor that interpolates the data, it is shown that the associated linear predictor $\beta_t = u_t \odot v_t$: (i) converges exponentially fast to zero outside of the support of $\beta^*$ (ii) is *with high probability* in a $\mathcal{O}(\sqrt{\eta\delta})$ neighborhood of $\beta^*$ in its support after a time $\mathcal{O}(\delta^{-1})$.

**Overall conclusion on the model.** During a first phase, SGD with large step sizes $\eta$ decreases the training loss until stabilization at some level set $\delta > 0$. During this loss stabilization, an effective noise-driven dynamics takes place. It shrinks the coordinates outside of the support of the sparsest signal and oscillates in parameter space at level $\mathcal{O}(\sqrt{\eta\delta})$ on its support. Hence, decreasing later the step size leads to perfect recovery of the sparsest predictor. This behaviour is illustrated in our experiments in Figure 2.

### 2.3.2   THE SPARSE FEATURE LEARNING CONJECTURE FOR MORE GENERAL MODELS

Results for diagonal linear nets recalled in the previous paragraph show that the noisy dynamics (9) induce a *sparsity bias*. As emphasized in HaoChen et al. (2021), this effect is largely due to the multiplicative structure of the noise $v \odot [X^\top \mathrm{d}B_t]$ that, in this case, has a shrinking effect *on the coordinates* (because of the coordinate-wise multiplication with $v$). In the general case, we see, thanks to Eq.(8), that the same multiplicative structure of the noise still happens but this time *with respect to the NTK feature matrix $\phi_\theta(X)$*. Hence, this suggests that similarly to the diagonal linear network case, the implicit bias of the noise can lead to a shrinkage effect applied to $\phi_\theta(X)$ which depends on the noise intensity $\delta$ and the step size of SGD. Indeed, an interesting property of Brownian motion is that, for $v \in \mathbb{R}^p$, $\langle v, B_t \rangle = \|v\|_2 W_t$, where the equality is valid in law and $(W_t)_{t\geq 0}$ is a one-dimensional Brownian motion. Hence, the process Eq.(8) is equivalent to a process whose $i$-th coordinate is driven by a noise proportional to $\|\phi_i\| \mathrm{d}W_t^i$, where $\phi_i$ is the $i$-th column of $\phi_\theta(X)$ and $(W_t^i)_{t\geq 0}$ is a one dimensional Brownian motion. This SDE structure, similar to the geometric Brownian motion, is expected to induce the shrinkage of each multiplicative factor (Oksendal, 2013, Section 5.1), i.e., in our case $(\|\nabla_\theta h(x_i)\|)_{i=1}^n$. Thus, we conjecture:

> *The noise part of Eq.(8) seeks to minimize the $\ell_2$-norm of the columns of $\phi_\theta(X)$.*

Note that the *fitting part* of the dynamics prevents the NTK feature matrix to collapse totally to zero, but as soon as they are not needed to fit the signal, *columns* can be reduced to zero. Remarkably, from a stability perspective, Blanc et al. (2020) showed a similar bias: locally around a minimum, the SGD dynamics implicitly tries to minimize the *Frobenius norm* $\|\phi_\theta(X)\|_\mathrm{F} = \sum_{i=1}^n \|\nabla_\theta h_\theta(x_i)\|^2$. We provide below a specification of this implicit bias for different architectures:

- **Diagonal linear networks:** For $h_{u,v}(x) = \langle u \odot v, x \rangle$, we have $\nabla_{u,v} h_{u,v}(x) = [v \odot x, u \odot x]$. Thus, for a generic data matrix $X$, minimizing the norm of each column of $\phi_{u,v}(X)$ amounts to put the maximum of zero coordinates and hence to minimize $\|u \odot v\|_0$.
- **ReLU networks:** We take the prototypical one hidden layer to exhibit the sparsification effect. Let $h_{a,W}(x) = \langle a, \sigma(Wx) \rangle$, then $\nabla_a h_{a,W}(x) = \sigma(Wx)$ and $\nabla_{w_j} h_{a,W}(x) = a_j x \mathbf{1}_{\langle w_j, x \rangle > 0}$. Note that the $\ell_2$-norm of the column corresponding to the neuron is reduced when it is activated at a *minimal number of training points*, hence the implicit bias enables the learning of *sparse data-active features*. Finally, when some directions are needed to fit the data, similarly activated neurons align to fit, allowing the rank of $\phi_\theta(X)$ to be also a good proxy for this feature sparsity.

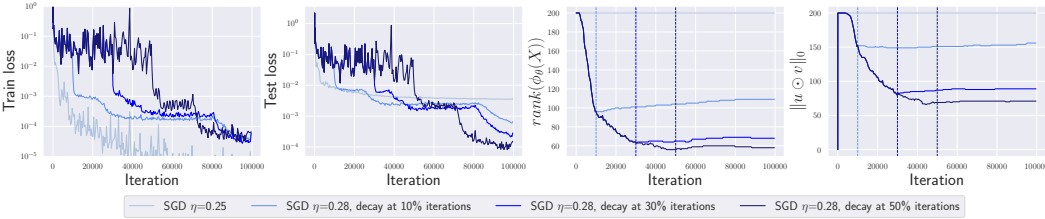

**Figure 2: Diagonal linear networks**. We observe loss stabilization, better generalization for longer schedules, minimization of the rank of $\phi_\theta(X)$ and sparsity of the predictor $u \odot v$.

Overall, fully understanding theoretically the structural implications of the implicit bias described above remains an exciting avenue for future work. We show next that the conjectured sparsity is indeed observed empirically for a variety of models, as well as that the rank reduction of $\phi_\theta(X)$ can be used as a good proxy of the hidden progress of the loss stabilization phase. This is confirmed both for SGD and its SDE modeling (the latter we show in App. C).

## 3 EMPIRICAL EVIDENCE OF SPARSE FEATURE LEARNING DRIVEN BY SGD

Here we present empirical resultsfor neural networks of increasing complexity: from diagonal linear networks to deep residual networks on CIFAR-10 and CIFAR-100. We make the following common observations for all these networks trained using SGD schedules with large step sizes:

(**O1**) **Loss stabilization**: training loss stabilizes around a high level set until step size is decayed,

(**O2**) **Generalization benefit**: longer loss stabilization leads to better generalization,

(**O3**) **Sparse feature learning**: longer loss stabilization leads to sparser features.

Importantly, *we use no explicit regularization* in our experiments so that the training dynamics is driven purely by SGD and the step size schedule. Additionally, in some cases, we cannot find a single large step size that would lead to loss stabilization. In such cases, whenever explicitly mentioned, we use a *warmup* step size schedule—i.e., increasing step sizes according to some schedule—to make sure that the training loss stabilizes around some level set. Such warmup schedules are commonly used in practice (He et al., 2016; Devlin et al., 2018). Warmup is often motivated purely from the optimization perspective as a way to accelerate training (Agarwal et al., 2021) but we suggest that, more importantly, it is also a way to amplify the regularization effect of the SGD noise which is proportional to the step size.

**Measuring sparse feature learning.** Our main insight is that the NTK feature matrix is significantly simplified in the loss stabilization phase, and that the rank of $\phi_\theta(X)$ (i.e., the sparsity of its singular values) is a good proxy to track this dynamics. We compute it over iterations for each model (except deep networks where it is not feasible) by using a fixed threshold on the singular values of $\phi_\theta(X)$ normalized by the largest singular value. In this way, we ensure that the difference in the rank that we detect is not simply due to a different scales of $\phi_\theta(X)$. Moreover, we always compute $\phi_\theta(X)$ on the number of fresh samples equal to the number of parameters $|\theta|$ to make sure that rank deficiency is not coming from $n \ll |\theta|$ which is the case in the overparametrized settings we consider.

Furthermore, we also want to track a more direct and interpretable notion of feature sparsity. This motivates us to count the average number of *distinct* (i.e., counting a group of highly correlated activations as one), *non-zero* activations at some layer over the training set which we refer to as the *feature sparsity coefficient*. We count a pair of activations $i$ and $j$ as highly correlated if their Pearson's correlation coefficient is at least 0.95. Unlike rank($\phi_\theta(X)$), the feature sparsity coefficient scales to deep networks and has an easy-to-grasp meaning.

### 3.1 SPARSE FEATURE LEARNING IN DIAGONAL LINEAR NETWORKS

**Setup.** The inputs $x_1, \ldots, x_n$ with $n = 80$ are sampled from $\mathcal{N}(0, \boldsymbol{I}_d)$ where $\boldsymbol{I}_d$ is an identity matrix with $d = 200$, and the outputs are generated as $y_i = \langle \beta_*, x_i \rangle$ where $\beta_* \in \mathbb{R}^d$ is $r = 20$ sparse. We

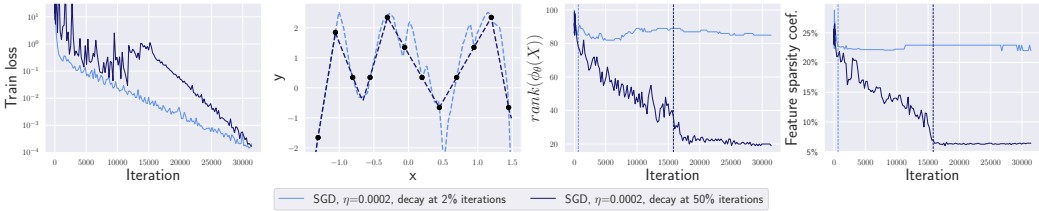

**Figure 4: Two-layer ReLU networks for 1D regression**. We observe loss stabilization, simplification of the model trained with a longer schedule, lower rank of $\phi_\theta(X)$, and much sparser features.

consider four different SGD runs (started from $u_i = 0.1$, $v_i = 0$ for each $i$): one with a small step size and three other with initial large step size decayed after $10\%$, $30\%$, $50\%$ iterations, respectively.

**Observations.** We show the results in Fig. 2 and note that **(O1)–(O3)** hold even in this simple model trained with vanilla SGD without any explicit regularization or layer normalization schemes. We observe that the training loss stabilizes around $10^{-1.5}$, the test loss improves for longer schedules, both rank($\phi_\theta(X)$) and $\|u \odot v\|_0$ decrease during the loss stabilization phase leading to a sparse final predictor. While the training loss has seemingly converged to $10^{-1.5}$, a hidden dynamics suggested by Eq.(9) occurs which slowly drifts the iterates to a sparse solution. This implicit sparsification explains the dependence of the final test loss on the time when the large step size is decayed, similarly to what has been observed for deep networks in Fig. 1. Interestingly, we also note that SGD with large step-size schedules encounters saddle points *after* we decay the step size (see the training loss curves in Fig. 2) which resembles the saddle-to-saddle regime described in Jacot et al. (2021) which does not occur in the large-initialization lazy training regime.

**SGD and GD have different implicit biases.** Since we observe from Fig. 2 that for loss stabilization, stochasticity alone does not suffice and large step sizes are necessary, one may wonder if conversely only large step sizes can be sufficient to have a sparsifying effect. Even if special instances can be found for which large step sizes are sufficient (such as for non-centered input features as in Nacson et al. (2022)), we answer this negatively showing that gradient descent in general does not go to the sparsest solution as demonstrated in Fig. 10 in the Appendix. Moreover, in Fig. 3, we visualize the difference in trajectory between the two methods taken with large step sizes over a 2D subspace spanned by $w^\star - w_{init}$ and $w_{flow} - w_{init}$,

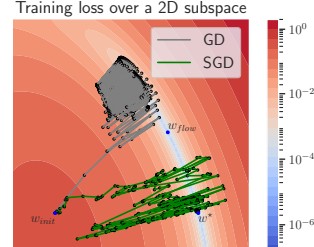

**Figure 3:** GD and SGD take different trajectories.

where $w^\star$ is the ground truth, $w_{flow}$ is the result of gradient flow, and $w_{init}$ is the initialization. This example provides an important intuition that loss stabilization alone is not sufficient for sparsification and that the role of noise described earlier is crucial.

## 3.2 SPARSE FEATURE LEARNING IN SIMPLE RELU NETWORKS

**Two-layer ReLU network in 1D.** We consider the one-dimensional regression task from Blanc et al. (2020) with 12 points, where label noise SGD has been shown to learn a simple model. We show that similar results can be achieved with large-step-size SGD via loss stabilization. We train a ReLU network with 100 neurons with SGD with a linear warmup (otherwise, we were unable to achieve approximate loss stabilization), directly followed by a step-size decay. The two plots correspond to a warmup/decay transition at $2\%$ and $50\%$ of iterations, respectively. The results shown in Fig. 4 confirm that **(O1)–(O3)** hold: the training loss stabilizes around $10^{-0.5}$, the predictor becomes much simpler and is expected to generalize better, and both rank($\phi_\theta(X)$) and the feature sparsity coefficient substantially decrease during the loss stabilization phase. For this one-dimensional task, we can directly observe the final predictor which is sparse in terms of the number of distinct ReLU kinks as captured by the feature sparsity coefficient and the rank of the NTK feature matrix. Interestingly, we also observed *overregularization* for even larger step sizes when we cannot fit all the training points (see Fig. 11 in Appendix). This phenomenon clearly illustrates how the capacity control is induced by the optimization algorithm: *the function class over which we optimize depends on the step size schedule*. Additionally, Fig. 12 in Appendix shows the evolution of the predictor over iterations. The

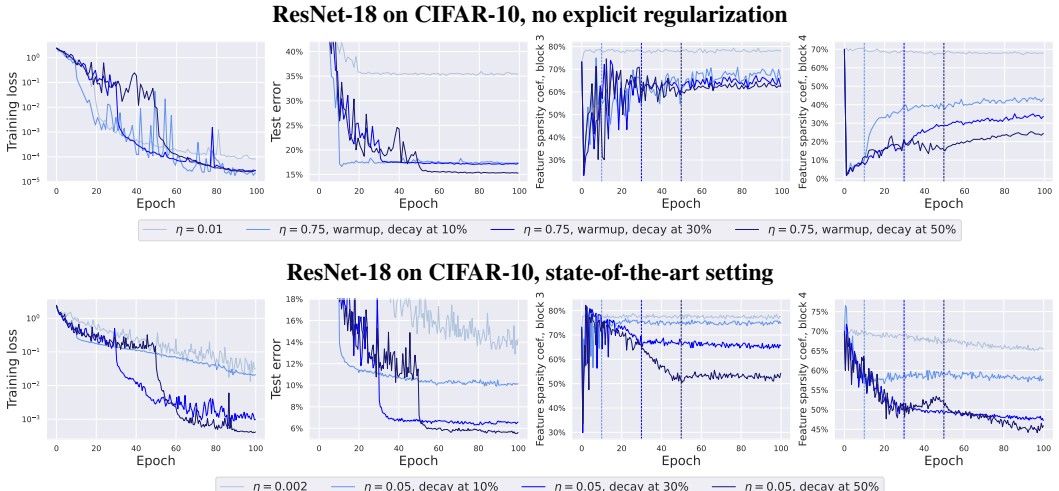

**Figure 5: Three-layer ReLU networks in a teacher-student setup**. We observe loss stabilization, lower rank of the NTK feature matrix and lower feature sparsity coefficient on *both* hidden layers.

**Figure 6: ResNet-18 trained on CIFAR-10.** Both *without explicit regularization* and in *the state-of-the-art setting*, the training loss stabilizes, the test loss noticeably depends on the length of the schedule, and the feature sparsity coefficient is minimized over iterations.

general picture is confirmed: first the model is simplified during the loss stabilization phase and only then fits the training data.

**Deeper ReLU networks.** We use a teacher-student setup with a random *three-layer* teacher ReLU network having 2 neurons on each hidden layer. The student network is overparametrized with 10 neurons on each layer and is trained on 50 examples. Such teacher-student setup is useful since we know that the student network can implement the ground truth function but might not find it due to the small sample size. We train models using SGD with a medium constant step size and a large step size with warmup decayed after $10\%, 30\%, 50\%$ iterations, respectively. The results shown in Fig. 5 confirm that **(O1)–(O3)** hold: the training loss stabilizes around $10^{-1.5}$, the test loss is smaller for longer schedules, and both $\text{rank}(\phi_\theta(X))$ and the feature sparsity coefficient substantially decrease during the loss stabilization phase. All methods have the same value of the training loss $(10^{-3})$ after $10^4$ iterations but different generalization. Moreover, we see that the feature sparsity coefficient decreases *on each layer* which makes this metric a promising one to consider for deeper networks.

### 3.3 SPARSE FEATURE LEARNING IN DEEP RELU NETWORKS

**Setup.** We consider here an image classification task and train a ResNet-18 and ResNet-34 on CIFAR-10 and CIFAR-100 using SGD with batch size 256 and different step size schedules. We use an exponentially increasing warmup schedule with exponent 1.05 to stabilize the training loss. We cannot measure the rank of $\phi(X)$ here since this matrix is too large ($\approx 50\,000 \times 20\,000\,000$) so we measure only the feature sparsity coefficient taken at two layers: at the end of super-block 3 (i.e., in the middle of the network) and super-block 4 (i.e., right before global average pooling at the end of the network) of ResNets. We test two settings: a basic setting without explicit regularizers and a state-of-the-art setting with weight decay, momentum, and standard augmentations.

**Observations.** The results on CIFAR-10 shown in Fig. 6 confirm that our main findings still hold also in this setting: the training loss stabilizes either slightly below $10^{-1}$ or above $10^{-1}$, the test error is becoming progressively better for longer schedules, as well as the feature sparsity coefficient. Small step sizes lead to bad generalization, especially without explicit regularization: $35\%$ test error compared to $15\%$ for large step sizes. This poor performance confirms that it is crucial to leverage the implicit bias of large step sizes. The difference in the feature sparsity coefficient is also substantial with the final model having $70\%$ instead of $24\%$ at block 4 without explicit regularization. The observations are similar for the state-of-the-art setting as well where even with explicit regularization, we still see a noticeable difference in generalization and feature sparsity depending on the step size and schedule. We further note that feature sparsity coefficient is gradually minimized over iterations in this case (similarly to Figures 2, 4, 5) while without explicit regularization we observe a different pattern: a very quick drop down to almost zero at the very first epoch and then a gradual increase.

We show the results with similar findings on CIFAR-100 in Fig. 15 in Appendix. Additionally, Fig. 14 illustrates that for small step sizes, the early and middle layers stay very close to their random initialization which indicates the absence of feature learning similarly to what is suggested by the neuron movement plot in Fig. 9 in the Appendix for two-layer network in a teacher-student setup.

## 4 INSIGHTS FROM OUR UNDERSTANDING OF THE TRAINING DYNAMICS

Here we provide an extended discussion on the implications of our theoretical and empirical findings.

**The multiple stages of the SGD training dynamics.** As analyzed and shown empirically, the training dynamics we considered can be split onto three distinct phases: (i) an initial phase of reducing the loss down to some level where stabilization can occur, (ii) a loss stabilization phase where noise and gradient directions combine to find architecture-dependent sparse representations of the data, (iii) a final phase when the step size is decreased to fit the training data. This typology allows to clearly disentangle the effect of the stabilization phase (ii) which relies on the implicit bias of SGD to simplify the model. Note that phases (ii) and (iii) can be repeated a few times until final convergence (He et al., 2016). Moreover, in some training schedules, (ii) does not explicitly occur, and the effect of loss stabilization (ii) and data fitting (iii) can occur simultaneously (Nakkiran et al., 2019).

**From lazy training to feature learning.** Similar sparse implicit biases have been shown for regression with infinitely small initialization (Boursier et al., 2022) and for classification (Chizat and Bach, 2020; Lyu and Li, 2020). However, both approaches are not practical from the computational point of view since (i) the origin is a saddle point for regression leading to the vanishing gradient problem (especially, for deep networks), and (ii) max-margin bias for classification is only expected to happen in the asymptotic phase (Moroshko et al., 2020). On the contrary, large step sizes enable to initialize far from the origin, while allowing to *efficiently* transition from a regime close to the lazy NTK regime (Jacot et al., 2018) to the rich feature learning regime.

**Common patterns in the existing techniques.** Tuning the step size to obtain loss stabilization can be difficult. To prevent early divergence caused by too large step sizes, we sometimes had to rely on an increasing step size schedule (known as *warmup*). Interpreting such schedules as a tool to favor implicit regularization provides a new explanation to their success and popularity. Additionally, normalization schemes like *batch normalization* or *weight decay*, beyond carrying their own implicit or explicit regularization properties, can be analyzed from a similar lens: they allow to use larger step sizes that boost further the implicit bias effect of SGD while preventing divergence (Bjorck et al., 2018; Zhang et al., 2018). Note also that we derived our analysis with batch size equal to one for the sake of clarity, but an arbitrary batch size $B$ would simply be equivalent to replacing $\gamma \leftarrow \gamma/B$. Similarly to the consequence of large step sizes, preferring *smaller batch sizes* (Keskar et al., 2016) while avoiding divergence seem key to benefit from the implicit bias of SGD. Finally, the effect of large step sizes or small batches is often connected to measures of *flatness* of the loss surface via stability analysis (Wu et al., 2018) and some methods like the Hessian regularization (Damian et al., 2021) or SAM (Foret et al., 2021) explicitly optimize it. Such methods resemble the implicit bias of SGD with loss stabilization implied by the label noise equation (Eq.(8)) where matrix $\phi_\theta(X)$ is the key component of the Hessian. However, an important practical difference is that the regularization strength in these methods is explicit and decoupled from the step size schedule which may be harder to properly tune since it is simultaneously responsible for optimization *and* generalization.

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

# APPENDIX

In Section A, we show Proposition 1 on the equivalence between SGD and GD with added noise. In Section B, we provide the proof that loss stabilization occurs as written in Proposition 2. In Section C, we show experimentally that the proposed SDE model matches well the SDE dynamics. Finally, we present additional experiments in Section D.

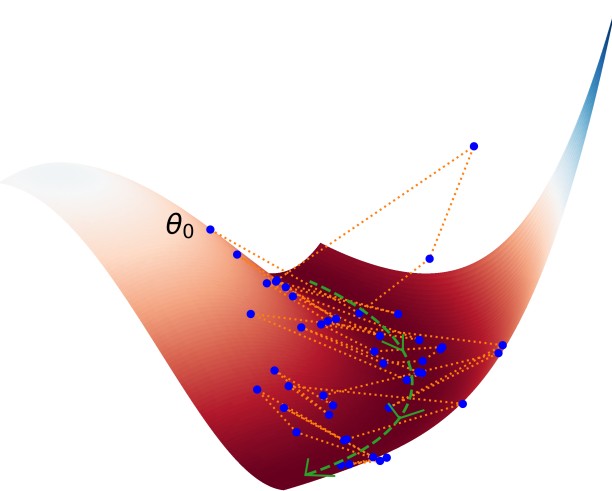

**Figure 7:** Three-dimensional visualisation of the SGD dynamics in a non-convex loss landscape. The SGD dynamics (blue points) is bouncing side-to-side to the bottom of the valley (the dotted green line). A slow movement occurs pushing the iterates in the direction given by the green arrows.

To begin this appendix, we provide in Figure 7 a toy visualization in which we showcase a typical SGD dynamics when loss stabilization occurs. We run SGD on the diagonal linear network with one sample in two dimensions ($n = 1, d = 2$) adding label noise of the shape given by equation Eq.(9), with balanced layers $u = v$. The blue points corresponds to iterates of the dynamics (that are linked with the orange dotted lines). The green line corresponds to the global minimum of the loss, what can be called the "bottom of the valley". This hopefully will serve the reader forge a visual intuition on (i) the bouncing dynamics side-to-side to the bottom of the valley (in green), and (ii) the slow stochastic movement (in the direction of the green arrows).

## A   SGD AND LABEL NOISE GD

For the sake of clarity we recall below the statement of the Proposition 1 which we prove in this section.

**Proposition 1.** *Let $(\theta_t)_{t \geq 0}$ follow the SGD dynamics Eq.(2) with sampling function $(i_t)_{t \geq 0}$. Let $\mathbf{1}_{i=i_t}$ be indicator function, define for $t \geq 0$, the random vector $\xi_t \in \mathbb{R}^n$ such that for all $i \in [\![1, n]\!]$,*

$$[\xi_t]_i := (h_{\theta_t}(x_i) - y_i)(1 - n\mathbf{1}_{i=i_t}). \tag{10}$$

*Then $(\theta_t)_{t \geq 0}$ follows the full-batch gradient dynamics on $\mathcal{L}$ with label noise $(\xi_t)_{t \geq 0}$, that is*

$$\theta_{t+1} = \theta_t - \frac{\eta}{n} \sum_{i=1}^{n} (h_{\theta_t}(x_i) - y_i^t) \nabla_\theta h_{\theta_t}(x_i), \tag{11}$$

*where we define the random labels $y^t := y + \xi_t$. Furthermore, $\xi_t$ is a mean zero random vector with variance such that $\frac{1}{n(n-1)} \mathbb{E} \|\xi_t\|^2 = 2\mathcal{L}(\theta_t)$.*

*Proof.* Note that

$$\sum_{i=1}^{n} (h_{\theta_t}(x_i) - y_i^t) \nabla_\theta h_{\theta_t}(x_i) = \sum_{i=1}^{n} (h_{\theta_t}(x_i) - y_i - [\xi_t]_i) \nabla_\theta h_{\theta_t}(x_i). \tag{12}$$

Using $[\xi_t]_i := (h_{\theta_t}(x_i) - y_i)(1 - n\mathbf{1}_{i=i_t})$,

$$= \frac{1}{n} \sum_{i=1}^{n} (h_{\theta_t}(x_i) - y_i - (h_{\theta_t}(x_i) - y_i)(1 - n\mathbf{1}_{i=i_t}))\nabla_\theta h_{\theta_t}(x_i), \tag{13}$$

$$= \sum_{i=1}^{n} \mathbf{1}_{i=i_t}(h_{\theta_t}(x_i) - y_i)\nabla_\theta h_{\theta_t}(x_i) = (h_{\theta_t}(x_{i_t}) - y_{i_t})\nabla_\theta h_{\theta_t}(x_{i_t}). \tag{14}$$

which is exactly the stochastic gradient wrt to sample $(x_{i_t}, y_{i_t})$.

Now we prove the latter part of the proposition regarding the scale of the noise. Recall that, for all $i \leqslant n$, we have $[\xi_t]_i = (h_{\theta_t}(x_i) - y_i)(1 - n\mathbf{1}_{i=i_t})$, where $i_t \sim \mathcal{U}(\llbracket 1, n \rrbracket)$. Now taking the expectation,

$$\mathbb{E}[\xi_t]_i = \mathbb{E}\left[(h_{\theta_t}(x_i) - y_i)(1 - n\mathbf{1}_{i=i_t})\right] = (h_{\theta_t}(x_i) - y_i)(1 - n\mathbb{E}\left[\mathbf{1}_{i=i_t}\right]) = 0, \tag{15}$$

as $\mathbb{E}\left[\mathbf{1}_{i=i_t}\right] = 1/n$. Coming to the variance,

$$\mathbb{E}\|\xi_t\|^2 = \mathbb{E}\left[\sum_{i=1}^{n} [\xi_t]_i^2\right] = \sum_{i=1}^{n} \mathbb{E}[\xi_t]_i^2 \tag{16}$$

$$= \sum_{i=1}^{n} (h_{\theta_t}(x_i) - y_i)^2 \mathbb{E}\left[(1 - n\mathbf{1}_{i=i_t})^2\right] \tag{17}$$

$$= \sum_{i=1}^{n} (h_{\theta_t}(x_i) - y_i)^2 \mathbb{E}\left[(1 - 2n\mathbf{1}_{i=i_t} + n^2\mathbf{1}_{i=i_t})\right] \tag{18}$$

$$= \sum_{i=1}^{n} (h_{\theta_t}(x_i) - y_i)^2 (1 - 2 + n) \tag{19}$$

$$= (n-1)\sum_{i=1}^{n} (h_{\theta_t}(x_i) - y_i)^2 = 2n(n-1)\mathcal{L}(\theta_t), - \tag{20}$$

and this concludes the proof of the proposition. $\square$

## B  QUADRATIC PARAMETERIZATION IN ONE DIMENSION

Again, for the Appendix to be self-contained, we recall the setup of the Proposition 2 on loss stabilization. We consider a regression problem with quadratic parameterization on one-dimensional data inputs $x_i$'s, coming from a distribution $\hat{\rho}$, and outputs generated by the linear model $y_i = x_i\theta_*^2$. The loss writes $F(\theta) := \frac{1}{4}\mathbb{E}_{\hat{\rho}}\left(y - x\theta^2\right)^2$, and the SGD iterates with step size $\eta > 0$ follow, for any $t \in \mathbb{N}$,

$$\theta_{t+1} = \theta_t + \eta\,\theta_t\,x_{i_t}\left(y_{i_t} - x_{i_t}\theta_t^2\right) \qquad \text{where} \quad x_{i_t} \sim \hat{\rho}. \tag{21}$$

We rewrite the proposition here.

**Proposition 3. (Extended version of Proposition 2)** Assume $\exists\, x_{\min}, x_{\max} > 0$ such that $\mathrm{supp}(\hat{\rho}) \subset [x_{\min}, x_{\max}]$. Then for any $\eta \in ((\theta_* x_{\min})^{-2}, 1.25(\theta_* x_{\max})^{-2})$, any initialization in $\theta_0 \in (0, \theta_*)$, for $t \in \mathbb{N}$, we have almost surely

$$F(\theta_t) \in \left(\epsilon_o^2\,\theta_*^2, 0.17\,\theta_*^2\right). \tag{22}$$

where $\epsilon_o = \min\left\{(\eta(\theta_* x_{\min})^2 - 1)/3, 0.02\right\}$. Also, almost surely, there exists $t, k > 0$ such that $\theta_{t+2k} \in (0.65\,\theta_*, (1 - \epsilon_o)\,\theta_*)$ and $\theta_{t+2k+1} \in ((1 + \epsilon_o)\,\theta_*, 1.162\,\theta_*)$.

*Proof.* Consider SGD recursion Eq.(21) and note that $y = x\theta_*^2$.

$$\theta_{t+1} = \theta_t + \eta\,\theta_t\,x(x\theta_*^2 - x\theta_t^2) \tag{23}$$

$$\theta_{t+1} = \theta_t + \eta\,\theta_t\,x^2\left(\theta_*^2 - \theta_t^2\right) \tag{24}$$

For the clarity of exposition, we consider the rescaled recursion of the original SGD recursion.

$$\theta_{t+1}/\theta_* = \theta_t/\theta_* + \eta\,\theta_*^2\,x^2\,\theta_t/\theta_*\left(1 - (\theta_t/\theta_*)^2\right), \tag{25}$$

and, by making the benign change $\theta_t \leftarrow \theta_t/\theta_*$, we focus on the stochastic recursion instead,

$$\theta_{t+1} = \theta_t + \gamma\theta_t(1 - \theta_t^2), \tag{26}$$

where $\gamma \sim \hat{\rho}_\gamma$ the pushforward of $\hat{\rho}$ under the application $z \to \eta\,\theta_*^2\,z^2$. Let $\Gamma := \text{supp}(\hat{\rho}_\gamma)$, the support of the distribution of $\gamma$. From the range of $\eta$, it can be verified that $\Gamma \subseteq (1, 1.25)$. Now the proof of the theorem follows from Lemma 5. $\qquad\square$

**Lemma 4** (Bounded Region). *Consider the recursion Eq.(26), for $\Gamma \subseteq (1, 1.25)$ and $0 < \theta_0 < 1$, then for all $t > 0$, $\theta_t \in (0, 1.162)$.*

*Proof.* Consider a single step of Eq.(26), for some $\gamma \in (1, 1.25)$,

$$\theta_+ = \theta + \gamma\theta(1 - \theta^2)$$

The aim is to show that $\theta_+$ stays in the interval $(0, 1.162)$. In order to show this, we do a casewise analysis.

For $\theta \in (0, 1]$: Since $0 < \theta \leq 1$, we have $\theta_+ \geq \theta > 0$. To prove the bound above, consider the following quantity,

$$\theta_{max} = \max_{\gamma\in(1,1.25)}\max_{\theta\in(0,1]} \theta + \gamma\theta(1 - \theta^2) \tag{27}$$

Say $h_\gamma(\theta) = \theta + \gamma\theta(1 - \theta^2)$, note that $h'_\gamma(\theta) = 1 + \gamma - 3\gamma\theta^2$ and $h''_\gamma(\theta) = -6\gamma\theta < 0$. Hence, for any $\gamma$ in our domain, the maximum is attained at $\theta_\gamma = \frac{1}{\sqrt{3}}\sqrt{\frac{1}{\gamma} + 1}$ and $h_\gamma(\theta_\gamma) = \frac{2(1+\gamma)^{3/2}}{3\sqrt{3\gamma}}$.

$$\max_{\gamma\in(1,1.25)}\max_{\theta\in(0,1]} \theta + \gamma\theta(1 - \theta^2) = \max_{\gamma\in(.5,1.25)} \frac{2(1+\gamma)^{3/2}}{3\sqrt{3\gamma}} \tag{28}$$

It can be verified that $\frac{2(1+\gamma)^{3/2}}{3\sqrt{3\gamma}}$ is increasing with gamma in the interval $(1, 1.25)$. Hence,

$$\max_{\gamma\in(1,1.25)} \frac{2(1+\gamma)^{3/2}}{3\sqrt{3\gamma}} \leq \left.\frac{2(1+\gamma)^{3/2}}{3\sqrt{3\gamma}}\right|_{\gamma=1.25} < 1.162 \tag{29}$$

Combining them, we get,

$$\theta_+ \leq \max_{\gamma\in(0,1.25)}\max_{\theta\in(0,1]} \theta + \gamma\theta(1 - \theta^2) < 1.162 \tag{30}$$

For $\theta \in (1, 1.162)$: Since $\theta > 1$, we have, $\theta_+ < \theta < 1.162$. For lower bound, note that for $\theta_+$ to be less than 0, we need $1 + \gamma - \gamma\theta^2 < 0$. But for $\gamma \in (1, 1.25)$ and $\theta \in (1, 1.162)$,

$$\gamma(\theta^2 - 1) < 1.25((1.162)^2 - 1) < 1. \tag{31}$$

Hence, it never goes below 0. $\qquad\square$

**Lemma 5.** *Consider the recursion Eq.(26) with $\Gamma \subseteq (1, 1.25)$ and $\theta_0$ initialized uniformly in $(0, 1)$. Then, there exists $\epsilon_0 > 0$, such that for all $\epsilon < \epsilon_0$ there exists $t > 0$ such that for any $k > 0$,*

$$\theta_{t+2k} \in (0.65, 1 - \epsilon) \quad and \quad \theta_{t+2k+1} \in (1 + \epsilon, 1.162) \tag{32}$$

*almost surely.*

*Proof.* Define $\gamma_{\min} > 1$ as the infimum of the support $\Gamma$. Let $\epsilon_o = \min\{(\gamma_{min}-1)/3, 0.02\}$. Note that $\epsilon_0 > 0$ as $\gamma_{\min} > 1$. Now for any $0 < \epsilon < \epsilon_o$, we have $\gamma_{\min}(2 - \epsilon)(1 - \epsilon) > 2$.

Divide the interval (0,1.162) into 4 regions, $I_0 = (0, 0.65]$, $I_1 = (0.65, 1 - \epsilon)$, $I_2 = [1 - \epsilon, 1)$, $I_3 = (1, 1.162)$. The strategy of the proof is that the iterates will eventually end up in $I_1$ and that once it ends up in $I_1$, it comes back to $I_1$ in 2 steps.

Let $\theta_0$ be initialized uniformly random in $(0, 1)$. Consider the sequence $(\theta_t)_{t\geq 0}$ generated by

$$\theta_{t+1} = h_{\gamma_t}(\theta_t) := \theta_t + \gamma_t\theta_t(1 - \theta_t^2) \quad \text{where} \quad \gamma_t \sim \hat{\rho}_\gamma. \tag{33}$$

We prove the following facts (**P1**)-(**P4**):

**(P1)** There exists $t \geq 0$ such that the $\theta_t \in I_1 \cup I_2 \cup I_3$.

**(P2)** Let $\theta_t \in I_3$, then $\theta_{t+1} \in I_1 \cup I_2$.

**(P3)** Let $\theta_t \in I_2$, there exists $k > 0$ such that for $k' < k$, $\theta_{t+2k'} \in I_2$ and $\theta_{t+2k} \in I_1$.

**(P4)** When $\theta_t \in I_1$, then for all $k \geq 0$, $\theta_{t+2k} \in I_1$ and $\theta_{t+2k+1} \in (1 + \epsilon, 1.162)$.

**Proof of (P1)-(P4)**: Let $t \in \mathbb{N}$, note first that the event $\{\theta_t = 1\} = \cup_{k \leqslant t} \{\theta_k = 1 | \theta_{k-1} \neq 1\}$ and hence a finite union of zero measure sets. Hence $\{\theta_t = 1\}$ is a zero measure set and therefore we do not consider it below. For any other sequence, from the above four properties, we can conclude that the lemma holds.

**Proof of P1**: Assume that until time $t > 0$, the iterates are all in $I_0$, then we have

$$\theta_t = \theta_{t-1}(1 + \gamma(1 - \theta_{t-1}^2)) \geq \theta_{t-1}(2 - \theta_{t-1}^2) > 1.5\,\theta_{t-1} > 1.5^t\,\theta_0 \tag{34}$$

Hence, the sequence eventually exits $I_0$. We know that it will stay bounded from Lemma 4, hence it will end up in $I_1 \cup I_2 \cup I_3$.

**Proof of P2**: For any $\theta_t \in (1, 1.162)$, $1 < \gamma < 1.25$, since $h_\gamma(.)$ is decreasing in $(1,1.162)$, we have $h_\gamma(1.162) < h_\gamma(\theta_t) < h_\gamma(1)$. Also $h_\gamma(\theta)$ is linear in gamma with negative coefficient for $\theta > 1$. Hence it decreases as $\gamma$ increases. Using this,

$$.652 = h_{1.25}(1.162) < h_\gamma(1.162) < h_\gamma(\theta_t) < h_\gamma(1) = 1. \tag{35}$$

Hence, $\theta_{t+1} \in I_1 \cup I_2$.

**Proof of P3**: The proof of this follows from Lemma 7.

**Proof of P4**: The proof of this follows from Lemma 10. $\qquad\square$

**Lemma 6.** *For any $\theta \in I_1 \cup I_2$ and any $a, b \in \Gamma$, $h_a(h_b(\theta)) \in I_1 \cup I_2$,*

$$h_{\gamma_{\max}}(h_{\gamma_{\max}}(\theta)) \leq h_a(h_b(\theta)) \leq h_{\gamma_{\min}}(h_{\gamma_{\min}}(\theta)). \tag{36}$$

*Proof.* For any $\gamma \in \Gamma$, recall

$$h_\gamma(\theta) = \theta + \gamma\theta(1 - \theta^2) = 1 + (1 - \theta)(\gamma\theta(1 + \theta) - 1). \tag{37}$$

Note that for $\theta \in I_1 \cup I_2$, $\theta(1 + \theta) > 1$, Hence $\gamma\theta(1 + \theta) > 1$. This gives us that $h_\gamma(\theta) > 1$. Now we will track where $\theta \in I_1 \cup I_2$ can end up after two stochastic gradient steps.

- For any $b \in \Gamma$, as $\theta \in I_1 \cup I_2$, we have

$$h_{\gamma_{\max}}(\theta) \geq h_b(\theta) \geq h_{\gamma_{\min}}(\theta) > 1,$$

  note $h_{\gamma_{\max}}(\theta) \geq h_b(\theta) \geq h_{\gamma_{\min}}(\theta)$ holds since $\theta < 1$.
- Now for any $a \in \Gamma$ and $x > 1$, $h_a(x)$ is a decreasing function in $x$. Hence

$$h_a(h_{\gamma_{\max}}(\theta)) \leq h_a(h_b(\theta)) \leq h_a(h_{\gamma_{\min}}(\theta)).$$

  Using $\gamma_{\min} \leq a$, $h_a(h_{\gamma_{\min}}(\theta)) \leq h_{\gamma_{\min}}(h_{\gamma_{\min}}(\theta))$, Similarly using $\gamma_{\max} > a$, we have, $h_{\gamma_{\max}}(h_{\gamma_{\max}}(\theta)) \leq h_a(h_{\gamma_{\max}}(\theta))$. Combining them we get,

$$h_{\gamma_{\max}}(h_{\gamma_{\max}}(\theta)) \leq h_a(h_b(\theta)) \leq h_{\gamma_{\min}}(h_{\gamma_{\min}}(\theta)). \tag{38}$$

Similar argument can extend it to,

$$h_{1.25}(h_{1.25}(\theta)) < h_a(h_b(\theta)) < h_1(h_1(\theta)). \tag{39}$$

$\qquad\square$

**Lemma 7.** *Let $\theta_t \in I_2$, there exists $k > 0$ such that $\theta_{t+2k} \in I_1$.*

*Proof.* For any $\gamma \in \Gamma$, let $\theta_+ = h_\gamma(\theta)$, then we have

$$h_\gamma(h_\gamma(\theta)) - \theta = h_\gamma(\theta_+) - \theta = \gamma\theta(1 - \theta^2) + \gamma\theta_+(1 - \theta_+^2). \tag{40}$$

Furthermore,

$$\theta_+ = \theta + \gamma\theta(1 - \theta^2) = \theta(1 + \gamma(1 - \theta^2)), \tag{41}$$

$$1 + \theta_+ = 1 + \theta + \gamma\theta(1 - \theta^2) = (1 + \theta)(1 + \gamma\theta(1 - \theta)), \tag{42}$$

$$1 - \theta_+ = 1 - \theta - \gamma\theta(1 - \theta^2) = (1 - \theta)(1 - \gamma\theta(1 + \theta)). \tag{43}$$

And multiplying the above three terms and adding $\theta(1 - \theta^2)$, we get,

$$\theta_+(1 - \theta_+^2) + \theta(1 - \theta^2) = \theta(1 - \theta^2)\{1 + \underbrace{\left[(1 + \gamma(1 - \theta^2))(1 + \gamma\theta(1 - \theta))(1 - \gamma\theta(1 + \theta))\right]}_{P(\theta)}\}$$

$$\tag{44}$$

For $\theta \in I_2$, using $\gamma_{\min}(2 - \epsilon)(1 - \epsilon) > 2$, we have the inequalities

$$(1 + \gamma(1 - \theta^2))(1 + \gamma\theta(1 - \theta)) > 1, \tag{45}$$

$$(1 - \gamma\theta(1 + \theta)) < 1 - \gamma_{min}(2 - \epsilon)(1 - \epsilon) < -1, \tag{46}$$

$$P(\theta) < -1. \tag{47}$$

Hence,

$$h_\gamma(h_\gamma(\theta)) - \theta = \gamma(1 - \theta^2)(1 + P(\theta)) < 0. \tag{48}$$

Therefore, for $[1 - \epsilon, 1)$, for any $\gamma \in \Gamma$, $h_\gamma(h_\gamma(\theta)) < \theta$. Hence for any two stochastic gradient step with $a, b \in \Gamma$, from Eq.(36), $\theta_{t+2} = h_a(h_b(\theta_t)) \leq h_{\gamma_{\min}}(h_{\gamma_{\min}}(\theta_t)) < \theta_t$. From any point in $I_2$, we have $|\theta_{t+2} - 1| > |\theta_t - 1|$, for any $a, b \in \Gamma$. Intutively this means that in *two gradient steps* the iterates move *further away from 1* until it eventually leaves the interval $I_2$ as the sequence $\{\theta_{t+2k}\}_{k \geq 0}$ is strictly decreasing with no limit point in $I_2$. From Lemma 9 , we know that in two steps the iterates will never leave $I_1 \cup I_2$. Hence they will eventually end up in $I_1$ leaving $I_2$. □

**Property 8.** *Define* $g_\gamma(\theta) := h_\gamma(h_\gamma(\theta))$ *for the sake of brevity. The followings properties hold for* $\theta \in I_1 \cup I_2$, $\gamma \in \Gamma$ *and* $\theta_\gamma$ *the root of* $h'_\gamma(\theta)$:

**Q1** $g_\gamma(\theta) \geq g_\gamma(\theta_\gamma)$.

**Q2** *The function* $g_\gamma(.)$ *is decreasing in* $[0.65, \theta_\gamma)$ *and increasing in* $(\theta_\gamma, 1]$.

*Proof.* Note $h'_\gamma(\theta) = 1 + \gamma - \gamma 3\theta^2$ has at most one root $\theta_\gamma \in (0, 1)$. Note that for all $\gamma \in \Gamma$, $\theta_\gamma \in I_1 \cup I_2$. For any $\gamma$, $g'_\gamma(\theta) = h'_\gamma(h_\gamma(\theta))h'_\gamma(\theta)$. For any $\theta \in I_1 \cup I_2$, we have, $h_\gamma(\theta) > 1 \implies h'_\gamma(h_\gamma(\theta)) < 0$. Therefore, $g'_\gamma(\theta)$ has only one root in $I_1 \cup I_2$. Since $\theta_\gamma \in I_1 \cup I_2$, note $g''_\gamma(\theta_\gamma) = h'_\gamma(h_\gamma(\theta_\gamma))h''_\gamma(\theta_\gamma) > 0$. Therefore, $g_\gamma(.)$ attains its minimum at $\theta_\gamma$ and this shows the desired properties. □

**Lemma 9.** *For any* $\theta \in I_1 \cup I_2$ *and any* $a, b \in \Gamma$, $h_a(h_b(\theta)) \in I_1 \cup I_2$.

*Proof.* **Lower Bound:** From Eq.(39), we know

$$h_{1.25}(h_{1.25}(\theta)) < h_a(h_b(\theta)) \tag{49}$$

We know that from property **Q1** that $g_\gamma(\theta) \geq g_\gamma(\theta_\gamma)$. Hence

$$g_{1.25}(\theta_{1.25}) < g_{1.25}(\theta) < h_a(h_b(\theta)) \tag{50}$$

It can be quickly checked that $.65 < g_{1.25}(\theta_{1.25})$. Hence the lower bound holds.

**Upper Bound:** From Eq.(39), we know

$$h_a(h_b(\theta)) < h_1(h_1(\theta)) \tag{51}$$

We know that from property **Q2** that $g_1(\theta) \leq \max\{g_1(1), g_1(0.65)\}$. It can be easily verified that $g_1(0.65) < 0.98$. Hence $g_1(\theta) < 1$.

□

**Lemma 10.** *For any $\theta \in I_1$ and any $a, b \in \Gamma$, $h_a(h_b(\theta)) \in I_1$ and $h_a(\theta) \in (1 + \epsilon, 1.162)$.*

*Proof.* The lower bound in Lemma 9 holds here. For the upper bound, from and Eq.(36),

$$h_a(h_b(\theta)) \leq h_{\gamma_{\min}}(h_{\gamma_{\min}}(\theta)). \tag{52}$$

Using property **Q2**,

$$h_{\gamma_{\min}}(h_{\gamma_{\min}}(\theta)) \leq \max\{g_{\gamma_{\min}}(1 - \epsilon), g_{\gamma_{\min}}(0.65)\} \tag{53}$$

From Eq.(48), $g_{\gamma_{\min}}(1 - \epsilon) < 1 - \epsilon$. From Eq.(39), $g_{\gamma_{\min}}(0.65) < g_1(0.65) < 0.98 < 1 - \epsilon$. In $I_1$, the function $h_a(.)$ first increases reaches maximum and decreases. Hence for $\theta \in I_1$, $h_a(\theta) \geq \min\{h_a(0.65), h_a(1 - \epsilon)\}$.

$$h_a(1 - \epsilon) \geq 1 - \epsilon + a(1 - (1 - \epsilon)^2)(1 - \epsilon), \tag{54}$$

$$= 1 - \epsilon + a(2\epsilon - \epsilon^2)(1 - \epsilon), \tag{55}$$

$$\geq 1 - \epsilon + \gamma_{\min}(2\epsilon - \epsilon^2)(1 - \epsilon), \tag{56}$$

$$= 1 + \epsilon + \epsilon\left(\gamma_{\min}(2 - \epsilon)(1 - \epsilon) - 2\right) > 1 + \epsilon. \tag{57}$$

Also $h_a(0.65) > h_1(0.65) > 1.02 > 1 + \epsilon$, therefore $h_a(\theta) > 1 + \epsilon$ and this completes the proof. $\square$

## C  EMPIRICAL VALIDATION OF THE SDE MODELING

In this section, we experimentally check the validity of the SDE modeling of SGD in Eq.(8) in terms of the key metrics: training loss, test loss, rank of the NTK feature matrix, and feature sparsity.

**SDE discretization.** Let $\gamma_t$ be the SDE discretization step size, $\eta_t$ the step size of the corresponding SGD that we aim to validate, $\delta_t$ the noise intensity level, and $Z_t \sim \mathcal{N}(0, I_n)$. Then we discretize the SDE from Eq.(8) as follows:

$$\theta_{t+1} = \theta_t - \gamma_t \nabla_\theta \mathcal{L}(\theta_t) + \sqrt{\gamma_t}\sqrt{\eta_t \delta_t}\, \phi_{\theta_t}(X)^\top Z_t. \tag{58}$$

To approximate continuous time, we use a small discretization step size $\gamma_t := \eta_t/10$ and run the discretization for $10\times$ longer than the corresponding SGD run. We use $\eta_t := \eta_{\lfloor t/10 \rfloor}^{SGD}$ and $\delta_t := c \cdot \mathcal{L}(\theta_{\lfloor t/10 \rfloor}^{SGD})$ where $c$ is a constant that we select for each setting separately to match the training dynamics of the corresponding SGD run. In addition, we also evaluate a discretization of gradient flow (i.e., Eq.(58) without the noise term) which helps to draw conclusions about the role of the noise term.

**Experimental results.** We present the discretization results in Fig. 8 for all models considered in the paper except deep networks for which computing the NTK matrix $\phi_{\theta_t}$ on each iteration of the SDE discretization is too costly. In all cases, the dynamics of the SDE discretization qualitatively matches the dynamics of the corresponding SGD run. In particular, we observe similar levels of decrease in the rank of the NTK matrix and feature sparsity coefficient. We note that the match between SDE and SGD curves is not expected to be precise due to the inherent randomness of the process. Finally, we observe that gradient flow discretization exhibits no rank minimization or feature sparsity which suggests that the presence of the noise (either from the original SGD or its SDE discretization) plays a key role in learning sparse features.

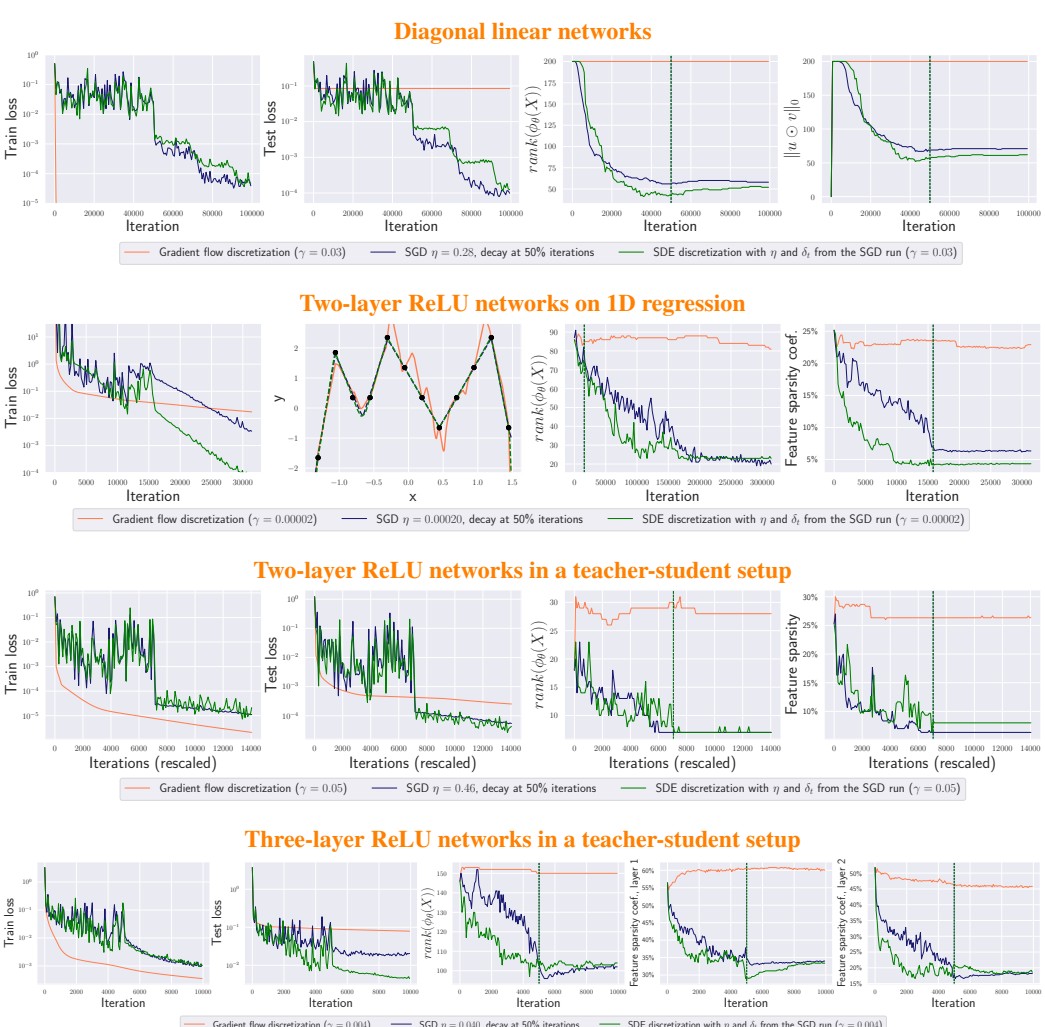

**Figure 8: Empirical validation of the SDE modeling.** In all cases, the dynamics of the SDE discretization qualitatively matches the dynamics of the corresponding SGD run. Moreover, gradient flow discretization exhibits no rank minimization or feature sparsity which suggests that the presence of the noise plays a key role in learning sparse features.

# D  ADDITIONAL EXPERIMENTAL RESULTS

This section of the appendix presents additional experiments complementing the ones presented in the main text.

**Illustration of neuron dynamics.** We illustrate the change of neurons during training of two-layer ReLU networks in the teacher-student setup of Chizat et al. (2019) (see Fig. 1 therein) using a large initialization scale for which small step sizes of GD or SGD lead to lazy training. We postpone the illustration of **(O1)–(O3)** to Fig. 13 in Appendix as our interest is on showing neuron dynamics (Fig. 9). We see that for SGD with a small step size, the neurons $w_i$ stay close to their initialization, while for a large step size, there is a clear clustering of directions $w_i$ along the teacher directions $w_i^\star$.

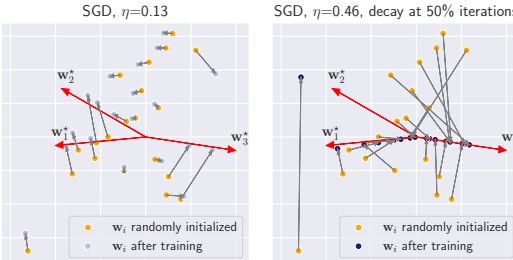

**Figure 9:** Only for a large step size, the neurons $w_i$ cluster along the teacher neurons $w_i^\star$ leading to a model that uses a sparse set of features.

The overall picture is very similar to Fig. 1 of Chizat et al. (2019) where the same feature learning effect is achieved via gradient flow from a small initialization which is, however, much more computationally expensive due to the saddle point at zero. Finally, we note that the clustering phenomenon of neurons $w_i$ motivates the removal of highly correlated activations in the feature sparsity coefficient: although the corresponding activations are often non-zero, many of them in fact implement *the same feature* and thus should be counted only once.

**Further results.** We give a short overview of additional figures referred to in the main text. More details can be found in the captions.

- Figure 10 shows that even if loss stabilization occurs in diagonal linear networks, the implicit bias towards sparsity is largely weaker than that of SGD and generalization is poor.
- Figures 11 and 12 demonstrate that the implicit bias resulting from high-loss stabilization makes the neural nets learn *first* a simple model *then eventually* fits the data.
- Figure 13 presents the sparsifying effect corresponding to the neurons' movements exhibited in Figure 9.
- Figure 14 showcases the features learning induced by large step sizes for different layers of ResNets-18 when trained on CIFAR-10.
- Figure 15 exhibits the feature sparsity in ResNets architecture on CIFAR-100 without any regularization (plain SGD) and in the state-of-the-art setup.

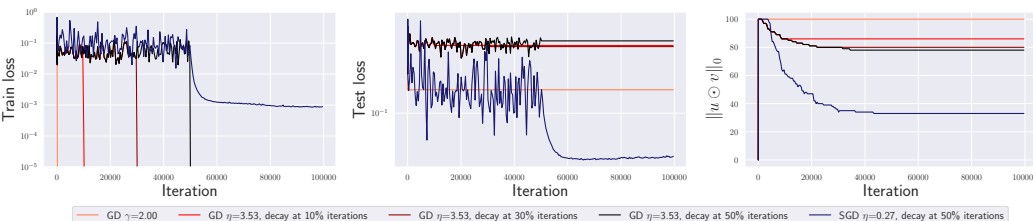

**Figure 10: Diagonal linear networks**. Loss stabilization also occurs for *full-batch gradient descent* but does not lead to a similar level of sparsity as SGD and also does not improve the test loss.

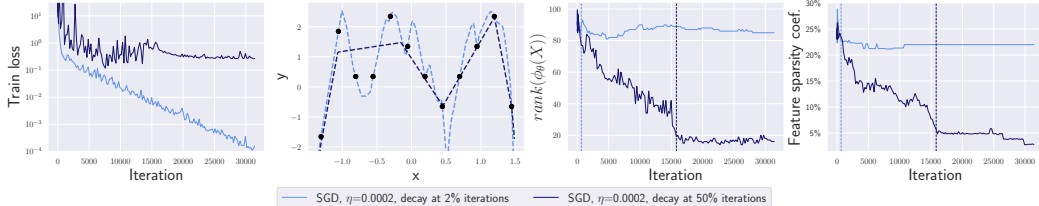

**Figure 11: Two-layer ReLU networks for 1D regression**. Unlike for Fig. 4, here we use a larger warmup coefficient ($500\times$ vs. $400\times$) which leads to overregularization such that the 50%-schedule run fails to fit all the training points and gets stuck at a too high value of the training loss ($\approx 10^{-0.5}$).

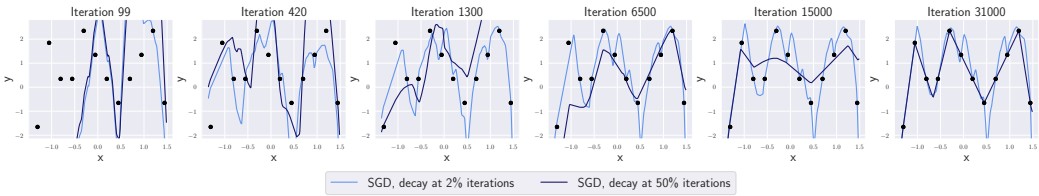

**Figure 12: Two-layer ReLU networks for 1D regression**. Illustration of the resulting models from Fig. 4 over training iterations. We can see that first the model is simplified and only then it fits the training data.

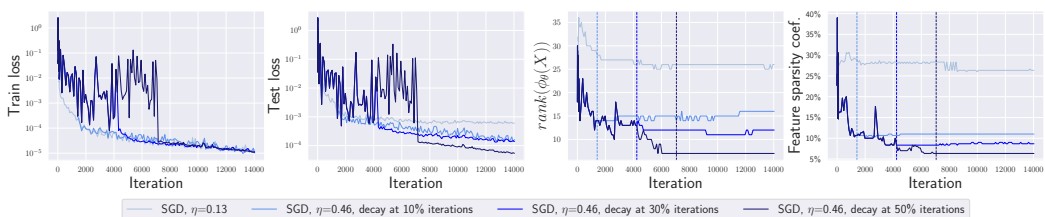

**Figure 13: Two-layer ReLU networks in a teacher-student setup**. Loss stabilization for two-layer ReLU nets in the teacher-student setup with input dimension $d = 2$. We observe loss stabilization, better test loss for longer schedules and sparser features due to simplification of $\phi(X)$.

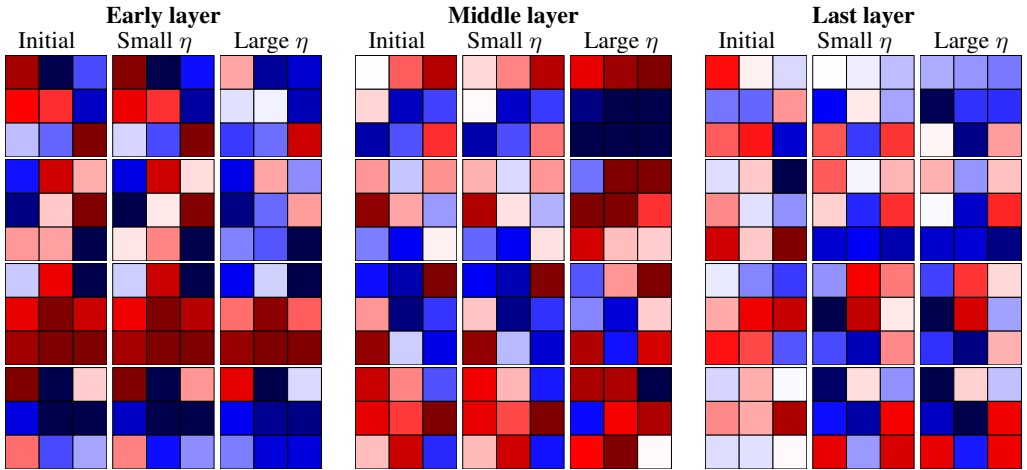

**Figure 14:** Visualization on four sets of convolutional filters taken from different layers of ResNets-18 trained on CIFAR-10 with small vs. large step size $\eta$ (the $50\%$ decay schedule). For small step sizes, the early and middle layers stay very close to randomly initialized ones which indicates the absence of feature learning.

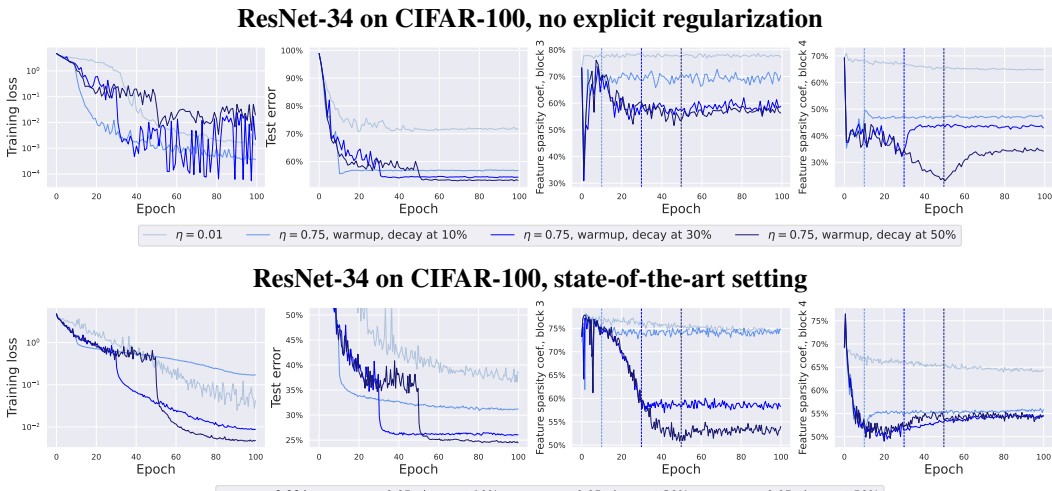

**Figure 15: ResNet-34 trained on CIFAR-100.** Both *without explicit regularization* and in *the state-of-the-art setting*, the training loss stabilizes, the test loss significantly depends on the length of the schedule, and feature sparsity is minimized over iterations. However, differently from the plots on CIFAR-10, here without explicit regularization we observe oscillating behavior after the step size decay (although at a very low level between $10^{-4}$ and $10^{-2}$).

