# OpenReview forum: "SGD with large step sizes learns sparse features"
_ICLR.cc/2023/Conference — Submitted to ICLR 2023_

### Official Review · Reviewer_FkNE · 2022-10-16

**Confidence:** 3
**Correctness:** 2
**Technical Novelty And Significance:** 2
**Empirical Novelty And Significance:** 3
**Recommendation:** 3

**Clarity, Quality, Novelty And Reproducibility:**

The novelty feels fine, even though some main contributions have been discovered here and there in the previous works

The main problem is clarity, and the overclaims I point out in the previous section

**Strength And Weaknesses:**

Strength: some pieces of insights are novel. For example, the authors suggest the mechanism of loss stabilization and suggest its connection to sparsity.

the point that GD and SGD have different inductive biases is also an important insight

The following are the weaknesses that the authors need to address.
1. the title is too strong: the title implies that using a large learning rate always leads to sparsity, but this is not what the paper has shown. The paper only showed that *empirically, a large learning rate leads to sparsity in certain restricted scenarios*. The title is a overclaim that needs to be changed

2. the first contribution has already been studied extensively in quite a few previous works. the authors need to discuss these works and modify the claim accordingly: (1) https://openreview.net/forum?id=uorVGbWV5sw; (2) https://proceedings.mlr.press/v162/mori22a.html

3. contribution 3: it is not exploration that leads to sparse features; it is the multiplicative noise. After all, "exploration" here is a very ambiguous word here and needs revision

4. page3: "we showcase our results for the mean square error, but other losses like the cross-entropy carry the same properties (W2021b)." I don't see which theorem in W2021b shows that cross-entropy carries the same properties. The authors should be more specific about which theorem in W2021b and which "property" they are referring to

5. prop 1: the notation $\mathbf{1}_{i=i_t}$ is undefined

6. prop 1 feels incorrect (it is, at least, unclear). The theorem statement seems to suggest that the SGD on $L(y)$ is equivalent to GD on $L(y + \xi)$, but this is not the case: the second equation of the proof does not compute the gradient with respect to $\xi_i$, which is also a function of $\theta$. The proof thus feels incorrect
- Thus, the message that SGD = GD + label noise is correct is also a misstatement.
- The authors needs to either correct the proof or update the proposition statement

7. the word "stabilize" is used many times but I feel is inappropriate. Look at figure 1: at a large learning rate, the loss does not stabilize to a fixed value but fluctuates drastically around a mean. I think the word "stabilize" is confusing and should be changed

8. the fonts in Figures 3, 4, 6, 7 are not visible to human eyes. The authors need to use a font size similar to the font size of the main text in the figures

9. the word "feature sparsity" is undefined throughout in the paper even though it plays an important role. this leads to a lot of potential confusion. For example, see the third figure in figure 7; the figure tells me that a "larger learning rate leads to smaller feature sparsity," but I guess this is not what the authors are trying to convey

10. I cannot agree with a main (empirical) claim of the paper: "longer stabilization leads to sparser features." The empirical results are only sufficient to show that longer stabilization is positively correlated with sparser features (because they happen simultaneously), but I do not think the empirical results allow one to conclude that one leads to the other. I stress the following point: *correlation cannot imply causality*




Minor:
1. perhaps the authors should point out that eq 2 is essentially SGD with batch size 1
2. the authors should give equation numbers to all the equations in the appendix so they are easier to refer to during the review

**Summary Of The Paper:**

The paper empirically shows that SGD with large step size learns sparse features and presented a few minimal theoretical examples that corroborate the empirical observation


**Summary Of The Review:**

The paper has great promise but is hindered by the disclarity, missing of discussion of important previous works, overclaim, and potential incorrectness of the theory

I cannot recommend the paper for acceptance in its current state. I might reconsider my position if the authors make extensive updates to improve the manuscript; otherwise, I suggest resubmitting to a different conference

---

> ### Author Response · Authors · 2022-11-12
> **Thanks for the detailed comments. We have made extensive updates to the paper. We answer the main concerns below. (Part 2)**
>
> **Prop 1 feels incorrect (it is, at least, unclear). The theorem statement seems to suggest that the SGD on L(y)  is equivalent to GD on  L(y+ξ), , but this is not the case: the second equation of the proof does not compute the gradient with respect to ξ, which is also a function of  θ. The proof thus feels incorrect. Thus, the message that SGD = GD + label noise is correct is also a misstatement. The authors need to either correct the proof or update the proposition statement.**
> We would like to emphasize that **our original Proposition 1 was correct**: in the proposition, we specifically said that **first** we take the gradient of the loss and **then** we inject the label noise. Mathematically this is saying that, if we consider the loss as a function of the data also: $L = L(\theta, x, y)$, then gradient descent with label noise $\xi_t$ corresponds to $\theta_{t+1} = \theta_t + \gamma \nabla_\theta L (\theta_t, x, y + \xi_t)$, where the gradient is taken only with respect to the **first variable**. We agree that we should have been more explicit about this point. We have updated the statement of Proposition 1 and written explicitly the effective label noise equation to avoid any confusion. Moreover, we have merged it with Lemma 2 for the sake of clarity. We hope the updated version resolves the issue.
>
> **The word "stabilize" is used many times but I feel is inappropriate. Look at figure 1: at a large learning rate, the loss does not stabilize to a fixed value but fluctuates drastically around a mean. I think the word "stabilize" is confusing and should be changed.** We used the word “stabilize” to indicate that the loss is neither diverging nor converging and can be found at an approximately constant level set. Since the relative magnitude of the oscillations is small with respect to the loss value, we feel the word stabilization is appropriate. However, we agree that it should not be understood as *stabilization to a fixed value*, thus we added in multiple places throughout the paper that loss stabilizes **on average**.
>
> **The word "feature sparsity" is undefined throughout in the paper even though it plays an important role. this leads to a lot of potential confusion.** Indeed, we agree that the original usage of “feature sparsity” could be confusing. Now throughout the paper, we refer specifically to the *“feature sparsity coefficient”* which was defined at the beginning of Section 3. Note we have now updated its description to be more specific. The *feature sparsity coefficient* corresponds to the average number of *distinct* (i.e., counting a group of highly correlated activations as one), *non-zero* activations at some layer *computed over the training set*. We count a pair of activations $i$ and $j$ as highly correlated if their Pearson's correlation coefficient is at least $0.95$. This is precisely what was measured in all the figures in Section 4.
>
> **I cannot agree with a main (empirical) claim of the paper: "longer stabilization leads to sparser features." The empirical results are only sufficient to show that longer stabilization is positively correlated with sparser features (because they happen simultaneously), but I do not think the empirical results allow one to conclude that one leads to the other. I stress the following point: correlation cannot imply causality.** We do not agree with this criticism. We note that our experimental setting is highly controlled: we increase the duration of loss stabilization (from 10% to 30% and 50% of the total number of iterations), and we observe progressively sparser features. Thus, we do not just measure two correlated variables, but instead **we perform a direct intervention on the duration of loss stabilization** and the sparser features appear **as a result of this intervention**. We think that it is a very convincing experimental setting to support our sparse feature learning conjecture. Moreover, there is more than only empirical evidence towards this conjecture: as mentioned in Section 2.3.1, the sparse feature learning claim has been **proven** for diagonal linear networks for label noise SGD. Given our modeling assumptions on standard SGD with loss stabilization, a similar mechanism is likely to be at play also for a generic case where the noise shrinks the columns of the NTK matrix instead of individual coordinates as in diagonal linear networks. We believe that a combination of the controlled experiments and theoretical results sufficiently justifies our sparse feature learning conjecture.
>
> **Prop 1: the notation $1_{i=i_t}$ is undefined.** Thanks, we have updated the paper.
>
> **The fonts in Figures 3, 4, 6, 7 are not visible to human eyes. The authors need to use a font size similar to the font size of the main text in the figures.** Thanks, we have modified the figures.

---

> > ### Comment · Reviewer_FkNE · 2022-11-15
> > **a quick additional review**
> >
> > I have not had the time to look at the reply in detail, but I want to follow up on a specific point while it is still possible for the authors to make updates -- The title still does not feel correct.
> >
> > While the authors argued that the current title is a valuable conjecture and is thus appropriate, I do not find it a good conjecture **because one can easily find counterexamples**. Let me submit a problem to the authors. I will accept the current title as appropriate if the authors can prove the title for the following minimal setting.
> >
> > Let $L(W) = \mathbb{E}\_x||Wx-y(x)||^2$ be the loss function of an arbitrary linear regression problem such that both the data covariance and $\mathbb{E}\_{x}[xy]$ are full-rank. For this loss function, can the authors prove that a large learning rate with SGD leads to the "features sparsity" they defined? Can the authors prove that a larger learning rate leads to a lower rank?
> >
> > If not (which I think is the case), the title is inappropriate because one definitely needs some additional conditions for the title to be sensible.

---

> > > ### Author Response · Authors · 2022-11-15
> > > **Thanks for the prompt reply. We agree that the sparsifying effect does not occur for least squares.**
> > >
> > > Thank you for your prompt reply.
> > >
> > > Our study focuses on neural networks and it is true that the sparsifying effect does not occur for least squares. Actually, note that when the parametrization is linear, the NTK features are **constant** so that there is no role of noise in this case. We will be clearer about this fact. It seems that it is not possible to change the title during the rebuttal period. However, if possible later and if the other reviewers and the AC are sharing the reviewer’s concerns, we will be happy to change the title to *"SGD with large step sizes **on neural networks typically** learns sparse features"*.

---

> > > > ### Comment · Reviewer_FkNE · 2022-11-15
> > > > **reply**
> > > >
> > > > Thanks for agreeing on the problem with the title.
> > > >
> > > > First of all, the title must be changed for me to recommend acceptance. While one cannot change the title of the openreview page, the authors can still change that in the manuscript, and I would like to see that.
> > > >
> > > > Second, the following sentence is not convincing: " if possible later and if the other reviewers **and** the AC are sharing the reviewer’s concerns, we will be happy to change the title to "SGD with large step sizes on neural networks typically learns sparse features." If we agree that there is a problem with the title, the authors should change it independent of the opinions of the other reviewers or AC. The authors need to acknowledge that there is a problem and promise a fix independent of other factors.

---

> > > > > ### Author Response · Authors · 2022-11-15
> > > > > **We think our current title is appropriate but we are open to change it if there is consensus among the reviewers**
> > > > >
> > > > > We understand your concern. However, we still think that our current title is appropriate for the reasons mentioned in our first reply. Moreover, the other reviewers have not pointed out any problem with the title. Thus, we would like to wait and have the advice and comments of the other reviewers on this fact.
> > > > >
> > > > > Finally, we note that the first sentence of the abstract is clear about the scope of our findings:
> > > > > *“We showcase important features of the dynamics of the Stochastic Gradient Descent (SGD) in the training of **neural networks**.“*

---

> ### Author Response · Authors · 2022-11-12
> **Thanks for the detailed comments. We have made extensive updates to the paper. We answer the main concerns below. (Part 1)**
>
> We thank the reviewer for the detailed comments. We have made an **extensive revision** of our manuscript improving the clarity and explanations based on the points brought up by the reviewer. In particular, we deemphasized the results from Section 2.1 and added the suggested reference of [Ziyin et al. (2022)](https://openreview.net/forum?id=uorVGbWV5sw). In addition, we note that **our original Proposition 1 is correct** but we have updated our statement to remove any potential confusion. In light of these clarifications, we hope the reviewer will consider reevaluation of our work.
>
> **The title is too strong: the title implies that using a large learning rate always leads to sparsity, but this is not what the paper has shown. The paper only showed that empirically, a large learning rate leads to sparsity in certain restricted scenarios. The title is a overclaim that needs to be changed.** We understand the concern of the reviewer, however, we believe that the title depicts well the message of our paper: SGD indeed learns sparse features when used with large step sizes. This message comes largely from the extensive *empirical* evidence that we present, and the abstract is very precise about this fact. We do not believe that asserting a strong idea—that we further discuss, conjecture from solid theory, and show experimentally—is implying any sort of overselling.
>
> **The first contribution has already been studied extensively in quite a few previous works. the authors need to discuss these works and modify the claim accordingly: (1) https://openreview.net/forum?id=uorVGbWV5sw; (2) https://proceedings.mlr.press/v162/mori22a.html.** Thank you very much for the interesting references we missed in our first version. Many articles study SGD with SDE as acknowledged throughout the paper and we do not claim to be the first to do so. Yet, we still believe that we are far from the end of the story when it comes to explaining the role of stochasticity in the performance of SGD. In particular, our main idea is to transfer the knowledge built on label-noise-driven dynamics to show how sparsity emerges due to the multiplicative noise of SGD. We believe that this idea offers quite a novel perspective on the topic.
>
> **Contribution 3: it is not exploration that leads to sparse features; it is the multiplicative noise. After all, "exploration" here is a very ambiguous word here and needs revision.** We want to avoid any source of confusion, thank you for pointing that out. The crux of our article lies in the multiplicative structure of the noise. However, note that the noise’s strength can vanish when the training loss goes to zero. This is precisely the reason why high-loss stabilization is beneficial: it helps to maintain high strength of the multiplicative noise. Hence, the aim of using the word “exploration” was to stress the fact that large step sizes allow the noise to act longer and more strongly. We admit that this may have been unclear, and we have rewritten all mentions of “exploration” in the paper.
>
> **Page 3: "we showcase our results for the mean square error, but other losses like the cross-entropy carry the same properties (W2021b)." I don't see which theorem in W2021b shows that cross-entropy carries the same properties. The authors should be more specific about which theorem in W2021b and which "property" they are referring to.** We apologize for the lack of precision: Lemma 2.14 of [W2021b](https://arxiv.org/abs/2105.01650) shows that the scaling of the noise covariance is also valid for the cross entropy loss. The structural property that the noise belongs to the linear spaces of the gradient is also valid for any type of loss. We have added a more specific reference to [W2021b](https://arxiv.org/abs/2105.01650) for the sake of completeness.

---

> > ### Comment · Reviewer_FkNE · 2022-11-15
> > **one more additional review**
> >
> > As I read through the paper a second time, I feel dubious about the following fact: all the training seems to proceed with weight decay.
> >
> > While the authors argued that this is the "standard practice," it becomes a problem particularly here because it has been shown that weight decay plus depth ($\geq 1$ hidden layers) is equivalent to a sparsity constraint and a low-rank constraint. See  https://www.stat.cmu.edu/~ryantibs/papers/sparsitynn.pdf
> >
> > This means that even if we just look at the minimizers of the training loss, a sparse / low-rank structure already emerges, and SGD having small or large step sizes may not really matter.
> >
> > Can the authors demonstrate the same phenomenon without weight decay? I understand the time constraint, but I would like to see at least one actually convincing example.

---

> > > ### Author Response · Authors · 2022-11-15
> > > **Almost all experiments are done without weight decay**
> > >
> > > The fact that sparsity occurs **without weight decay** is the crux of our message. Hence, almost (see below) all experiments are done **without weight decay**.  Furthermore, at the beginning of Section 3, we explicitly write: *“Importantly, we use no explicit regularization in our experiments so that the training dynamics is driven purely by SGD and the step size schedule.”*
> > >
> > > We used weight decay exceptionally in two cases:
> > > - Figure 1 to represent a typical training curve obtained in practice for ResNets on CIFAR-10.
> > > - Figure 6 (bottom row) and Figure 15 (bottom row) to represent a state-of-the-art setting on CIFAR-10 / CIFAR-100 with data augmentation, momentum, and weight decay.
> > >
> > > All other figures in the paper including Figures 2, 3, 4, 5, 6 (top row) in the main part were obtained **without weight decay**.

---

> > > > ### Comment · Reviewer_FkNE · 2022-11-15
> > > > **reply**
> > > >
> > > > Thanks for the clarification, and I apologize for my neglect. Let me think more.

---

> ### Author Response · Authors · 2022-11-17
> **Further discussion**
>
> We thank again the reviewer for all the comments. Since the deadline of the public discussion is approaching (tomorrow, November 18), we would like to kindly ask to let us know if our rebuttal has addressed the reviewer's concerns. We have made extensive updates to the paper based on the reviewer's suggestions. And, as mentioned in the other comment, we are ready to change the title if the other reviewers agree.

---

### Official Review · Reviewer_gDbe · 2022-10-24

**Confidence:** 4
**Clarity, Quality, Novelty And Reproducibility:** This is generally well-written work o…
**Correctness:** 2
**Technical Novelty And Significance:** 2
**Empirical Novelty And Significance:** 3
**Recommendation:** 5

**Strength And Weaknesses:**

The paper is generally well-written. Understanding the implicit bias of SGD with large step sizes is a significant theoretical and practical issue. This paper provides a rich discussion and offers a sparse feature learning perspective.  However, I feel that neither the analysis is well-designed nor the claim is well-supported. Specifically, I have two primary concerns from the main text of this work: the persuasiveness of theories in Section 2 and the sparse feature bias of deep networks in Section 3.
These two main points also limit the significance or rather limit my ability to determine this work's significance.


- The persuasiveness of theories in Section 2.
    - **The reasonableness of the SDE in (Eqn. 7).**  SDE is good modeling of SGD only in small step size regime, but it is generally unclear how this SDE modeling is relevant for understanding SGD with large step sizes.  As the authors mentioned in Section 2.3.1, this SDE may be reasonable for the diagonal linear model. However, for nonlinear networks such as ReLU nets, the reasonableness of the SDE modeling (eqn 7) seems questionable (especially the noise covariance). This SDE is used to motivate the bias of learning sparse features but is never used in experiments, where vanilla SGD is applied. It is thus unclear why do not use vanilla SGD to motivate the sparse feature learning.   The authors can address this issue using experiments.

    - **The hidden stochastic dynamics orthogonal to the bouncing directions.** In the abstract, the authors mention ''this loss stabilization induces a hidden stochastic dynamics orthogonal to the bouncing directions''. However, it seems that this paper does not point out the bouncing direction as well as explain the reason for orthogonality. The 1-D example (Eqn. 4) gives us insights into loss stabilization, but it cannot help us understand the authors' claim about the bouncing direction and the orthogonal hidden stochastic dynamics. The authors can construct a 3-D toy example to illustrate the loss stabilization, the bouncing direction, and the orthogonal effective dynamics.

    - **The claim that large step sizes learn sparse feature solutions.**
    I do not follow the explanation in Section 2.3.2 of why the eqn (7) implies that SGD implicitly minimizes the $\ell_2$ norm of each column of the feature matrix. This claim is so big and concrete but I do not see too much support for this claim. In addition, all of the authors' analysis holds for any step size in the interval of step sizes that satisfy loss stabilization. It means that SGD with different step sizes in this interval has the same sparse-feature-learning bias. Hence, it seems that the authors' viewpoint can not distinguish the different implicit biases of two sizes in this interval. If this is the case I do not view it as grounds for rejection but it is important for a reader to know.


- The sparse feature bias of deep networks in Section 3.

    - **The clarification of the feature sparsity coefficient.** What is the feature sparsity coefficient for deep neural networks mentioned on Page 6? As an essential metric, there is neither a clear explanation nor a mathematical formulation of this in the main text, which is not very clear.



**Summary Of The Paper:**

This paper studies the implicit bias of SGD with large step sizes. The authors present empirical observations and some theoretical understanding. They show that large step sizes SGD enables exploration by loss stabilization, and this exploration leads SGD to learn sparse features. The authors also provide some insights into the deep neural networks' training dynamics and understanding of some existing tricks.

**Summary Of The Review:**

This paper contains very interesting observations and analyses. However, I feel that the results are overclaimed and analysis support is not enough. Consequently,  I am a bit skeptical about the applicability of the findings.

---

> ### Author Response · Authors · 2022-11-12
> **Thanks for the very insightful and precise questions (particularly about SDE modeling!) that helped us to improve the manuscript.**
>
> We thank the reviewer for the assessment of our work and the very insightful and precise questions. The questions helped us make our main messages more concrete, and improve the manuscript. We agree that our SDE modeling needed to be empirically evaluated and **we provide an extensive list of experiments validating our SDE modeling** in Appendix C. We also provide a toy 3D visualization demonstrating our formulation of bouncing and slow stochastic dynamics. We provide a detailed answer to the questions below.
>
> **SDE is a good modeling of SGD only in small step size regime, but it is generally unclear how this SDE modeling is relevant for understanding SGD with large step sizes. Critical issues with ReLU.** The reviewer is correct on this fact: all *consistency* theorems that can be proved belong to the infinitesimally small step size regime. However, we argue the following:
> 1. the large step sizes effect is well modelled by the loss stabilization, and **once we observe and plug this property into the equation** we believe that the SDE given in Eq.(8) makes sense to model the large step size SGD dynamics
> 2. the SDE shall not be seen as a strict approximation of the SGD dynamics but more like a model to understand its main characteristics, in the spirit of why historically SDEs have been built for (see e.g. [1])
> 3. more importantly, as the reviewer wisely suggested, **we decided to present an extensive set of experiments in Appendix C showing that the SDE model and SGD with large step size match qualitatively** (and even quantitatively to some extent!). We thank the reviewer for this suggestion as we believe that it strongly strengthens our SDE study to understand the SGD dynamics.
>
> **However, it seems that this paper does not point out the bouncing direction as well as explain the reason for orthogonality. Toy model is required to understand.** The reviewer is correct: this part was unclear and without enough space to discuss this in-depth, we decided to remove these assertions. Note that we added, in the Appendix (Figure 7), a 3D visualization of the dynamics in the loss landscape for a toy model that hopefully will help the reader forge an intuition on the separation between the bouncing direction and the slow stochastic dynamics.
>
> **I do not follow the explanation in Section 2.3.2 of why the eqn (7) implies that SGD implicitly minimizes the ℓ2-norm of each column of the feature matrix. This claim is so big and concrete but I do not see too much support for this claim.** We apologize if this paragraph was unclear. This claim comes from standard studies of the geometric Brownian motion (e.g. see [2, Chapter 5]). In a word, this prototypical stochastic process exemplifies how the multiplicative noise induces a shrinkage effect *per coordinate*. Since each coordinate corresponds (in law) to a one-dimensional Brownian motion time the norm of each column, we conjecture that the shrinkage effect should apply to the columns of the multiplicative noise matrix. We rephrased this part and added references to help the reader understand the role of multiplicative noise in stochastic processes.
>
> **SGD with different step sizes in this interval has the same sparse-feature-learning bias. Hence, it seems that the authors' viewpoint can not distinguish the different implicit biases of two sizes in this interval.** Thanks for this interesting remark! *Qualitatively*, the reviewer is correct: there shall be no important difference between two step sizes whenever they both induce (high) loss stabilization. *However*, the discussion of the diagonal linear case, section 2.3.1 shows that the story may be more subtle: both the time scale of convergence and the concentration of the weights near convergence depend on the step size (through $\eta$ and $\delta$). More concretely: the larger the step size, the higher the loss stabilizes and the faster (and stronger) the sparsifying effect will take place. We have added this discussion to the text.
>
> **The clarification of the feature sparsity coefficient.** Indeed, we agree that the original description was not precise. We have revised that part of the paper to be more specific. The **feature sparsity coefficient** is the average number of *distinct* (i.e., counting a group of highly correlated activations as one), *non-zero* activations at some layer *computed over the training set*. We count a pair of activations $i$ and $j$ as highly correlated if their Pearson's correlation coefficient is at least $0.95$. Importantly, the feature sparsity coefficient, unlike $\text{rank}(\phi_\theta(X))$, scales to deep networks and has an easy-to-grasp meaning.
>
> [1] Stochastic Stability of Differential Equations (Stochastic Modelling and Applied Probability, 66), R. Khasminskii, 1980.
>
> [2] Stochastic Differential Equations: An Introduction with Applications. B. Oksendal, 2003.

---

> ### Author Response · Authors · 2022-11-17
> **Further discussion**
>
> We thank again the reviewer for all the comments. Since the deadline of the public discussion is approaching (tomorrow, November 18), we would like to kindly ask to let us know if our rebuttal has addressed the reviewer's concerns. We believe that we have addressed the main points (and revised the paper accordingly), particularly about the reasonableness of the SDE modeling. We have also provided more details on the sparse feature learning claim stemming from the SDE and on the feature sparsity coefficient that we have used for the experiments.

---

### Official Review · Reviewer_Jpgc · 2022-10-24

**Confidence:** 4
**Correctness:** 4
**Technical Novelty And Significance:** 3
**Empirical Novelty And Significance:** 3
**Recommendation:** 8

**Clarity, Quality, Novelty And Reproducibility:**

The paper is overall clearly-written but needs to be polished to improve the readability. The idea is quite novel. The authors don’t provide the code and hence it’s not clear if all of the results are reproducible.


**Strength And Weaknesses:**

### Strength:
* The idea is novel and interesting.
* The authors provide some synthetic examples to interpret the intuition.
* The authors conduct lots of experiments to demonstrate the conjecture.

### Weakness:
* It’s a little bit hard to follow as the authors don’t provide clear logic in the presentation of Section 2.
* The examples are a little bit toy.


**Summary Of The Paper:**

This paper provides some insights on the potential reason that the initial large step sizes can learn the model that can generalize well in practice. Center to the claim is an SDE with specific noise covariance structure, and the authors argue that, if we use the large step sizes at the beginning, the loss should stabilize at a certain level without further decreasing. During this phase, the unnecessary feature will be optimized towards 0, which will eventually lead to a solution with sparse features.


**Summary Of The Review:**

The authors provide a quite good intuition to interpret the effectiveness of large initial step sizes. Although the example is a little bit toy, I believe this paper will make impacts in the related areas. I feel the authors should be happy to include some discussions for [1]. Although it appears after the ICLR submission deadline, I feel this paper provides some theoretical analysis that potentially demonstrates the intuition discussed in this paper.

[1] Mousavi-Hosseini, Alireza, Sejun Park, Manuela Girotti, Ioannis Mitliagkas, and Murat A. Erdogdu. "Neural Networks Efficiently Learn Low-Dimensional Representations with SGD." arXiv preprint arXiv:2209.14863 (2022).

---

> ### Author Response · Authors · 2022-11-12
> **Thanks for the positive assessment. We have improved readability of the paper and included the code as supplementary material.**
>
> We thank the reviewer for the positive assessment of our work. We have updated the paper to improve its readability and also **included the code** as supplementary material. We provide detailed answers to the comments below.
>
> **It’s a little bit hard to follow as the authors don’t provide clear logic in the presentation of Section 2.** We revised this section to (i) clarify the specificity of SGD’s noise, (ii) emphasize the possibility of loss stabilization (iii) discuss more deeply the multiplicative effect of the noise which leads to sparsity.
>
> **Only toy examples.** We believe our models are simple but effective. We reproduce our results across a variety of models of increasing complexity. Note that the simplest model in our paper, i.e., the diagonal neural network already exhibits a rich behaviour in terms of implicit regularization and has garnered a lot of attention in recent years ([Woodworth et al, 2020](https://arxiv.org/abs/2002.09277), [HaoChen et al., 2020](https://arxiv.org/abs/2006.08680), [Pesme et al., 2021](https://arxiv.org/abs/2106.09524)). Furthermore, we reproduce the same behaviour across various models indicating that the phenomenon generalizes to deep networks used in practice (such as ResNets).
>
> **Comparison to [1] Mousavi-Hosseini, Alireza, Sejun Park, Manuela Girotti, Ioannis Mitliagkas, and Murat A. Erdogdu. "Neural Networks Efficiently Learn Low-Dimensional Representations with SGD." arXiv preprint arXiv:2209.14863 (2022).** Thanks for pointing out this very interesting paper. The result states that online SGD with *weight decay* learns low-dimensional representation *in the mean-field limit* when *initialization is small*, in the same vein as Chizat and Bach (2020) and Woodworth et al. (2020).  Instead, in our work, we use a standard initialization (of larger scale) and we try to show that the noise of SGD together with large step sizes helps transition from the kernel (large initialization) to the rich regime. The fact that these particular limits recover similar features as ours is truly remarkable.

---

> ### Author Response · Authors · 2022-11-17
> **Further discussion**
>
> We thank again the reviewer for the positive feedback. Since the deadline of the public discussion is approaching (tomorrow, November 18), we would like to kindly ask to let us know if there are additional concerns that we could address.

---

### Official Review · Reviewer_USkr · 2022-10-28

**Confidence:** 3
**Correctness:** 3
**Technical Novelty And Significance:** 3
**Empirical Novelty And Significance:** 2
**Recommendation:** 8

**Clarity, Quality, Novelty And Reproducibility:**

Besides the caveats mentioned above, the authors argumentation are not hard to follow, and their findings are of sufficient novelty and interest. The authors seem to prevent enought details with respect to their experimentation to reproduce their results.

Below are some more minor points for the authors to either help with their exposition or writing.

- Pg. 1 end of Par. 1: sentence unclear.
- Pg. 1 Par.3: Which noise are the authors addressing?
- Pg. 2 Par. 1: In what way "noise covariance [is] being rarely well understood"?
- Pg. 2 Par. 2: "Loss stabilization" is an important concept for this paper but is not explicitly defined early on.
- Pg. 2 Par. 2: Definition of fast-slow dynamics, at least a reference to the larger literature would be helfpul.
- Pg. 2 Par. 3: What does saddle-to-saddle vs. side-to-side mean?
- Pg. 2 Par. 3: leads to learn -> leads the model to learn
- Pg. 2 Par. 4: Insights from our research?
- Pg. 2 Par. 4: "Fitting phase" is not introduced. Neither is "exploration" is properly defined.
- Pg. 2 Par. 6: Why are such definitions "questionable"?
- Pg. 3 Par. 2: The authors intended meaning "''hidden' dynamics" should be explained as early as possible.
- Pg. 3 Par. 2: Which parts of the entire training dynamics is not captured by these analyses?
- Pg. 3 Par. 3: What's the "central phase" of a training.
- Pg. 3 Par. 4: Can the authors qualify / demonstrate which subset of their analyses directly translate to other types of losses?
- Pg. 3 Par. 4: train loss -> training loss.
- Pg. 3 Par. 5: "We note that..." can the authors support this assertion?
- Pg. 3: Instead of "specific label noise", I'd recommend the authors name the particular noise they are proposing and use that term.
- Pg. 3 Proposition 1: Let ... follows -> Let ... follow (also in the appendix)
- Pg. 3 Proposition 1: Please fix the numberings of the propositions etc. in the appendix.
- Pg. 4 Par 2: Please provide a reference for the convergence results mentioned.
- Pg. 5 Par 1: Sparse-inducing -> sparsity inducing.
- Pg. 5 Par 2: ... example of simple non-linear networks ... are diagonal linear networks?
- Pg. 6 Par 3: How do the authors interpret the warm up to serve implicit regularization - their exposition before does not seem to refer to any such function of learning rate increase during early training.
- Pg. 8 Par 1: Why do the authors use the teacher-student setting for illustrating their method.

**Strength And Weaknesses:**

The authors' argumentation seems to include the following steps:

1- SGD can be formulated as GD with a specific random noise (Prop 1.).
2- Variance of the norm of the specific random noise they introduced is proportional to the batch loss (Lemma 2.).
3- It is possible for an SGD trained algorithm to maintain an equilibrium where the loss is lower and upper bounded all the while the parameters take different values in the basin of the minimum but at some distance away from it (Prop 3.).
4- In the phase of the learning where the loss is constant, SGD can be modelled by a SDE whose noise term is characterized by the stochastic gradients (Eq 7).
5- This SDE implies that when the loss is constant, the optimization procedure will be inclined to reduce the l_2 norm of the stochastic gradients, leading to a sparse representation.

I find the authors' work valuable in investigating how the training dynamics and sparse representations evolve during training of neural networks. How efficient representations are achieved throughout the training procedure is a topic that is promising for understanding how neural networks generalize. The authors have a more or less self-consistent argumentation (albeit with the caveats below) and provide interesting experimental results to complement their claims.

Although I believe that the conference audience would benefit from their work, I believe answering the questions below would improve my and possibly other conference audience's understanding of their work:

- The authors' comparison of their approach to the rest of the literature is cursory at best. Although they mention a lot of other works, they usually shy away from making comparisons that illuminate why their results differed from those previously published. I would appreciate even less citations in exchange for meaningful comparison. For example, (Pg. 2 Par 5) What makes authors experimental regime different than that of Lewkowycz et al. 2020.? What does flatness definitions being "questionable" mean (Pg. 2)? Why are their findings regarding batch GD is different than those of Nacson et al 2022? (Pg 7.) In which aspects are authors' demonstrations are "closer to practice" (Pg. 1) and compared to whom?
- Pg. 4 Par. 4: How would the authors argue that Proposition 3 supports the importance of large learning rates per se in leading to the loss stabiliziation? Moreover, how does Prop. 1 serves the authors' exposition exactly? What in framing SGD as a GD + noise enable Claim 4 to be made, that would not be possible if it was not framed as GD?
- What is the justification for the choice of Brownian motion in the authors' proposed SDE? Large learning rates are assumed to produce heavy-tailed behavior in neural network parameters, would these results change the interpretation of the authors' results?
- Throughout the introduction and exposition the authors refer to label noise being a key in understanding the generalization behavior of SGD. I think this might be confusing for some audience since the type of label noise the authors are proposing is simply a theoretical device for their derivation, as opposed to researchers working with uncertainty in supervision signal.

**Summary Of The Paper:**

The authors claim that in SGD-trained neural networks, during the phases where the loss is constant, the stochastic noise in the optimization serves to allow the model to learn sparse features, which endows it with a benevolent inductive bias in terms of generalization. The authors demonstrate their ideas in experiments with small to medium sized neural networks.

**Summary Of The Review:**

I think the authors' work provides some interesting findings that would be helpful for a more nuanced understanding of the generalization behavior of deep neural networks. At points, more justification for their modeling choices and/or their inferences seem to be needed, as well as more concrete comparions with the existing body of work.

---

> ### Author Response · Authors · 2022-11-12
> **Thanks for your detailed feedback. We answer the secondary points below. (Part 2)**
>
> Now we discuss the remaining points.
>
> **Noise covariance [is] being rarely well understood?** The true covariance structure of SGD’s noise is very intricate, hence, we want to say that oversimplified setups are usually considered (e.g., identity noise covariance). We want to emphasize that we keep almost all of its complexity in this article—which is rarely done.
>
> **Saddle-to-saddle vs. side-to-side mean?** Saddle-to-saddle means that the dynamics go from one saddle point to the other: in dynamical systems, these dynamics are called *heteroclinic lines* and are the subject of intense studies. Side-to-side refers to the fact that there is (at least) one direction that is “bouncing” above a valley, as shown in Proposition 3. We have added a toy 3D visualization at the beginning of the Appendix to help the reader forge an intuition.
>
> **What's the "central phase" of a training?** A typical training loss curve observed in practice (e.g., [He et al, 2016](https://arxiv.org/abs/1512.03385)) goes as follows: a first sudden drop (early phase), then a long phase during which the loss oscillates but stays approximately constant on average (**this is what we call the central phase**), and finally the convergence phase, where the training loss drops significantly. We have added a brief clarification.
>
> **Can the authors qualify/demonstrate which subset of their analyses directly translate to other types of losses?** Similar calculations (e.g., also made in Lemmas 2.14 and 2.16 of [4]) show that the SDE model would be the same with the cross-entropy. Then, *the same sparsifying conjecture should apply*. We restricted ourselves to the square loss for the sake of concreteness.
>
> **We note that..." can the authors support this assertion? (about mini-batch B>1)** Taking a mini-batch of size $B>1$ would simply multiply the noise covariance by 1/B. The rest of our analysis will be unchanged.
>
> **How do the authors interpret the warm up to serve implicit regularization?** We interpret warmup as a way to stabilize the loss, i.e., to make sure that it does not converge or diverge. The implicit regularization benefit thus comes from the effect of the SGD noise during the loss stabilization phase.
>
>
> [4] Stochastic gradient descent with noise of machine learning type PART I: discrete time analysis. S. Wojtowytsch., 2021.

---

> ### Author Response · Authors · 2022-11-12
> **Thanks for your detailed feedback. We answer the main points below. (Part 1)**
>
> We thank the reviewer for the positive assessment and detailed feedback. We agree with the comments about the related work, Propositions 1 and 3, the need to discuss the SDE modeling, and the numerous writing suggestions. We have updated the paper accordingly. We discuss the most important points below.
>
> **“The authors' comparison of their approach to the rest of the literature is cursory at best.”**
> - **Lewkowycz et al. 2020 (Pg. 2).** The catapult mechanism of [Lewkowycz et al. (2020)](https://arxiv.org/abs/2003.02218) describes a different training regime where the training loss shows only one spike at the start of training and then monotonically converges *without loss stabilization*. Moreover, their analysis considers full-batch *GD* whereas our work aims at understanding the effect of large step sizes on *SGD*. The implicit regularization effect proposed therein is also different from ours. We have been clearer about the difference in our revision.
> - **Flatness definitions being "questionable" mean (Pg. 2)**
> Many typically used flatness definitions are questionable in distinguishing the generalization of different minima since (i) they are not invariant under reparametrizations that lead to an equivalent neural network [(Dinh et al. (2017))](https://arxiv.org/abs/1703.04933) and (ii) even for naturally trained networks, full-batch gradient descent with large step sizes (unlike SGD) can lead to flat solutions which are not well-generalizing [(Kaur et al. (2022))](https://arxiv.org/abs/2206.10654).
> - **Nacson et al 2022? (Pg 7.)**  Nacson et al. (2022) study full-batch **GD** instead and report improvements in generalization for large step sizes *only* on highly uncentered data for diagonal nets. This is in contrast to our study of large step size **SGD** which leads to better generalization even for centered data.
> - **"Closer to practice" (Pg. 1) and compared to whom?**
> By “closer to practice” we meant considering the training schedules from the [ResNet paper](https://arxiv.org/abs/1512.03385) which are widely used in practice. This is in comparison to the theoretical works whose results are only expected to hold asymptotically with a very slow convergence rate or for infinitesimally small step sizes. We modified that part to remove the potential confusion.
>
> **Why Proposition 3 supports the importance of large learning rates?** We believe Proposition 3 does support well the importance of a large learning rate since it indeed assumes a **lower bound** on the step size. Furthermore, this lower bound on the step size is **necessary** for loss stabilization: the loss provably converges to zero for smaller step sizes. We emphasized this fact in the revised version.
>
> **How does Prop. 1 serve the authors' exposition exactly?** The purpose of this proposition is twofold: (i) it gives immediately two important pieces of information about the noise structure induced by SGD; it belongs to the linear space spanned by the gradients and it gives the scale of the stochastic noise (proportional to the loss itself), (ii) it allows to transfer knowledge from label noise dynamics into studying our general framework. This is the starting point of the analysis and hence, we have emphasized these facts in the revised version. Thanks for the pointer.
>
> **Brownian motion or heavy tails.** The hypothesis of finite-variance noise has indeed been contested in a series of works by Gurbuzbalaban and coauthors (e.g. [1]), paving the way for heavy-tail diffusions. Yet, there have been many doubts in the literature that this atypical behavior happens in practice, and **both empirical and theoretical studies have proved the converse** (see e.g. the rigorous study in [2] contesting the experiments of [1] and also [3]). Hence, as finite-variance noise leads to Brownian-driven SDE, we focused on this modeling. Furthermore, as an important empirical validation, note that we complemented our paper with SDE experiments that match the dynamics of SGD (see Appendix C), showing the consistency of our approach.
>
> **Label noise is only a theoretical vision and not really what the authors do.** Thanks for the meaningful comment. As said above, the label noise reformulation is indeed a theoretical device to better understand the geometry of the noise induced by SGD; we revised the manuscript to be clearer on this point.
>
>
> [1] The Heavy-Tail Phenomenon in SGD, M. Gurbuzbalaban, U. Simsekli, L. Zhu,  2021
>
> [2] On the Validity of Modeling SGD with Stochastic Differential Equations (SDEs). Z. Li, S. Malladi, S. Arora, 2021.
>
> [3] Non-Gaussianity of Stochastic Gradient Noise. A. Panigrahi, R. Somani, N. Goyal, P. Netrapalli, 2019.

---

> ### Author Response · Authors · 2022-11-17
> **Further discussion**
>
> We thank again the reviewer for all the comments. Since the deadline of the public discussion is approaching (tomorrow, November 18), we would like to kindly ask to let us know if our rebuttal has addressed the reviewer's concerns. We believe that we have addressed the main points (and revised the paper accordingly) about related work, Propositions 1 and 3 (Propositions 1 and 2 in the revised version), and heavy-tailed noise. We have also incorporated the numerous suggestions to improve the exposition and writing.

---

> > ### Comment · Reviewer_USkr · 2022-11-21
> > **Thanks**
> >
> > I thank the authors for their response and find that the changes they made in the main text improved the comprehensibility of their paper considerably. I changed my score to reflect the authors' improvements.

---

### Author Response · Authors · 2022-11-12
**A summary of the main changes**

We thank all the reviewers for their feedback and their many constructive suggestions.

We have revised the paper based on the reviewers’ suggestions where the main changes are highlighted in orange. Here is the list of the most important updates:
- We have added experiments on the validity of our SDE modeling (Appendix C). We emphasize that we see a very good match between large step-size SGD and the SDE discretization in terms of key metrics (training loss, test loss, rank of the NTK feature matrix, and feature sparsity coefficient).
- We have clarified the definition of the *feature sparsity coefficient* we used throughout the paper (see the beginning of Section 3).
- We have made our claims and contributions more specific (Section 1.1). In particular, we have deemphasized the results from Section 2.1. These results are necessary for our exposition but we agree they were rather expected given the previous works of [HaoChen et al., 2021](https://arxiv.org/abs/2006.08680) and [Ziyin et al., 2022](https://arxiv.org/abs/2102.05375).
- We have clarified Proposition 1 and merged it with Lemma 2 to be more concise (we note that this change shifted the ordering of the subsequent propositions and equations).
- We have explained better our focus on label noise SGD.
- The related work is covered in more detail from now on.
- We have implemented numerous other suggestions that improve clarity and readability.
- Finally, we have added the code for reproducibility purposes.

---

### Decision · Program_Chairs · 2023-01-20

**Decision:**

Reject

**Justification For Why Not Higher Score:**

The main claims in the paper may not have been demonstrated in a sufficiently convincing manner. In particular, following the discussion phase, there are still lingering doubts regarding whether a causal link has been established between large learning rates and the post-training sparsity.

**Justification For Why Not Lower Score:**

N/A.

**Metareview: Summary, Strengths And Weaknesses:**

This paper examines SGD and argues empirically and theoretically through a connection to an SDE that large steps sizes can induce sparse features.

The reviewers generally thought that the connection to the SDE was a bit tenuous, and even with the addition of further validating experiments there were still split opinions about whether the theoretical contributions were nevertheless useful: some reviewers thought they provided a nice plausibility argument, while others viewed them as a stretch and a bit of a distraction. Ultimately, the theoretical results are not in themselves strong enough to form the backbone of the paper.

The empirical analysis was the subject of extended discussion. A main concern was that the experimentation was narrow, and it was not necessarily demonstrated that the mechanism behind the effect was in fact the same across the different settings. All together, there were some doubts about the main conclusions. I encourage the authors to develop more convincing evidence before resubmitting to another venue.

**Summary Of Ac-Reviewer Meeting:**

The AC-reviewer meeting highlighted a number of new concerns with the paper that did not come across clearly in the written discussion phase.

* None of the reviewers were convinced by the theoretical analysis via the SDE, though there was some disagreement about whether they were useful as plausibility arguments.
* The reviewers did not fully understand the motivation for or definitions of the sparsity metric.
* The toy experiments (diagonal linear net) were unconvincing
* The causal relationship between step size and sparsity was not well established.